# Understanding Dimensional Collapse in Contrastive Self-supervised Learning

**Li Jing, Pascal Vincent, Yann LeCun, Yuandong Tian**
Facebook AI Research
{ljng, pascal, yann, yuandong}@fb.com

## Abstract

Self-supervised visual representation learning aims to learn useful representations without relying on human annotations. Joint embedding approach bases on maximizing the agreement between embedding vectors from different views of the same image. Various methods have been proposed to solve the collapsing problem where all embedding vectors collapse to a trivial constant solution. Among these methods, contrastive learning prevents collapse via negative sample pairs. It has been shown that non-contrastive methods suffer from a lesser collapse problem of a different nature: dimensional collapse, whereby the embedding vectors end up spanning a lower-dimensional subspace instead of the entire available embedding space. Here, we show that dimensional collapse also happens in contrastive learning. In this paper, we shed light on the dynamics at play in contrastive learning that leads to dimensional collapse. Inspired by our theory, we propose a novel contrastive learning method, called *DirectCLR*, which directly optimizes the representation space without relying on a trainable projector. Experiments show that *DirectCLR* outperforms SimCLR with a trainable linear projector on ImageNet.

## 1 Introduction

Self-supervised learning aims to learn useful representations of the input data without relying on human annotations. Recent advances in self-supervised visual representation learning based on joint embedding methods (Misra & Maaten, 2020b; He et al., 2020; Chen et al., 2020a; Chen & He, 2020; Grill et al., 2020; Zbontar et al., 2021; Bardes et al., 2021; Chen et al., 2020b; Dwibedi et al., 2021; Li et al., 2021; Misra & Maaten, 2020a; HaoChen et al., 2021; Assran et al., 2021; Caron et al., 2021) show that self-supervised representations have competitive performances compared with supervised ones. These methods generally aim to learn representations invariant to data augmentations by maximizing the agreement between embedding vectors from different distortions of the same images.

As there are trivial solutions where the model maps all input to the same constant vector, known as the collapsing problem, various methods have been proposed to solve this problem that rely on different mechanisms. Contrastive methods like Chen et al. (2020a) and He et al. (2016) define 'positive' and 'negative' sample pairs which are treated differently in the loss function. Non-contrastive methbods like Grill et al. (2020) and Chen & He (2020) use stop-gradient, and an extra predictor to prevent collapse without negative pairs; Caron et al. (2018; 2020) use an additional clustering step; and Zbontar et al. (2021) minimize the redundant information between two branches.

These self-supervised learning methods are successful in preventing complete collapse whereby all representation vectors shrink into a single point. However, it has been observed empirically in non-contrastive learning methods (Hua et al., 2021; Tian et al., 2021) that while embedding vectors do not completely collapse; they collapse along certain dimensions. This is known as *dimensional collapse* (Hua et al., 2021), whereby the embedding vectors only span a lower-dimensional subspace.

In contrastive methods that explicitly use positive and negative pairs in the loss function, it seems intuitive to speculate that the repulsive effect of negative examples should prevent this kind of dimensional collapse and make full use of all dimensions. However, contrary to intuition, contrastive learning methods still suffer from dimensional collapse (See Fig. 7). In this work, we theoretically study the dynamics behind this phenomenon. We show there are two different mechanisms that

cause collapsing: **(1)** along the feature direction where the variance caused by the data augmentation is larger than the variance caused by the data distribution, the weight collapses. Moreover, **(2)** even if the covariance of data augmentation has a smaller magnitude than the data variance along all dimensions, the weight will still collapse due to the interplay of weight matrices at different layers known as implicit regularization. This kind of collapsing happens only in networks where the network has more than one layer.

Inspired by our theory, we propose a novel contrastive learning method, called *DirectCLR*, which directly optimizes the encoder (i.e., representation space) without relying on a trainable projector. *DirectCLR* outperforms SimCLR with a linear trainable projector on ImageNet.

We summarize our contributions as follows:

- We empirically show that contrastive self-supervised learning suffers from dimensional collapse whereby all the embedding vectors fall into a lower-dimensional subspace instead of the entire available embedding space.
- We showed that there are two mechanisms causing the dimensional collapse in contrastive learning: (1) strong augmentation along feature dimensions (2) implicit regularization driving models toward low-rank solutions.
- We propose *DirectCLR*, a novel contrastive learning method that directly optimizes the representation space without relying on a trainable projector. *DirectCLR* outperforms SimCLR with a linear trainable projector.

## 2    RELATED WORKS

**Self-supervised Learning Methods** Joint embedding methods are a promising approach in self-supervised learning, whose principle is to match the embedding vectors of augmented views of a training instance. Contrastive methods (Chen et al., 2020a; He et al., 2016) directly compare training samples by effectively viewing each sample as its own class, typically based on the InfoNCE contrastive loss (van den Oord et al., 2018) which encourages representations from positive pairs of examples to be close in the embedding space while representations from negative pairs are pushed away from each other. In practice, contrastive methods are known to require a large number of negative samples. Non-contrastive methods do not directly rely on explicit negative samples. These include clustering-based methods (Caron et al., 2018; 2020), redundancy reduction methods (Zbontar et al., 2021; Bardes et al., 2021) and methods using special architecture design (Grill et al., 2020; Chen & He, 2020).

**Theoretical Understanding of Self-supervised Learning** Although self-supervised learning models have shown success in learning useful representations and have outperformed their supervised counterpart in several downstream transfer learning benchmarks (Chen et al., 2020a), the underlying dynamics of these methods remains somewhat mysterious and poorly understood. Several theoretical works have attempted to understand it. Arora et al. (2019b); Lee et al. (2020); Tosh et al. (2021) theoretically proved that the learned representations via contrastive learning are useful for downstream tasks. Tian et al. (2021) explained why non-contrastive learning methods like BYOL (Grill et al., 2020) and SimSiam (Chen & He, 2020) work: the dynamics of the alignment of eigenspaces between the predictor and its input correlation matrix play a key role in preventing complete collapse.

**Implicit Regularization** It has been theoretically explained that gradient descent will drive adjacent matrices aligned in a linear neural network setting (Ji & Telgarsky, 2019). Under the aligned matrix assumption, Gunasekar et al. (2018) prove that gradient descent can derive minimal nuclear norm solution. Arora et al. (2019a) extend this concept to the deep linear network case by theoretically and empirically demonstrating that a deep linear network can derive low-rank solutions. In general, over-parametrized neural networks tend to find flatter local minima (Saxe et al., 2019; Neyshabur et al., 2019; Soudry et al., 2018; Barrett & Dherin, 2021).

## 3    DIMENSIONAL COLLAPSE

Self-supervised learning methods learn useful representation by minimizing the distances between embedding vectors from augmented images (Figure 1a). On its own, this would result in a collapsed

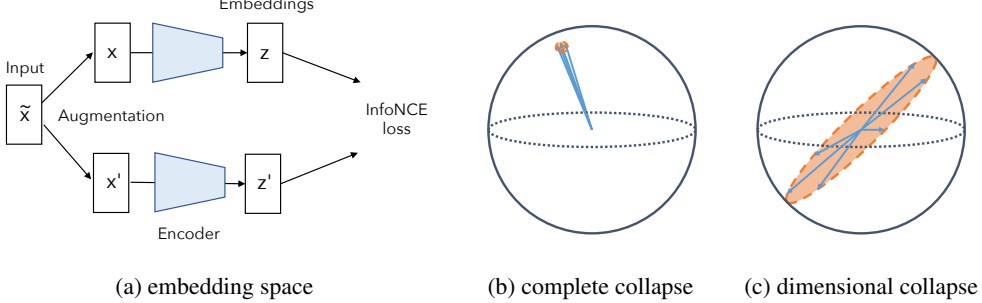

(a) embedding space      (b) complete collapse      (c) dimensional collapse

Figure 1: Illustration of the collapsing problem. For complete collapse, the embedding vectors collapse to same point. For dimensional collapse, the embedding vectors only span a lower dimensional space.

solution where the produced representation becomes constant (Figure 1b). Contrastive methods prevent complete collapse via the negative term that pushes embedding vectors of different input images away from each other. In this section, we show that while they prevent complete collapse, contrastive methods still experience a dimensional collapse in which the embedding vectors occupy a lower-dimensional subspace than their dimension (Figure 1c).

We train a SimCLR model (Chen et al. (2020a)) with a two-layer MLP projector. We followed the standard recipe and trained the model on ImageNet for 100 epoch. We evaluate the dimensionality by collecting the embedding vectors on the validation set. Each embedding vector has a size of $d = 128$. We compute the *covariance matrix* $C \in \mathbb{R}^{d \times d}$ of the embedding layer (here $\bar{\mathbf{z}} := \sum_{i=1}^{N} \mathbf{z}_i / N$ and $N$ is the total number of samples):

$$C = \frac{1}{N} \sum_{i=1}^{N} (\mathbf{z}_i - \bar{\mathbf{z}})(\mathbf{z}_i - \bar{\mathbf{z}})^T \quad (1)$$

Figure 2 shows singular value decomposition on this matrix ($C = USV^T, S = diag(\sigma^k)$). in sorted order and logarithmic scale ($\{\log(\sigma^k)\}$). We observe that a number of singular values collapse to zero, thus representing collapsed dimensions.

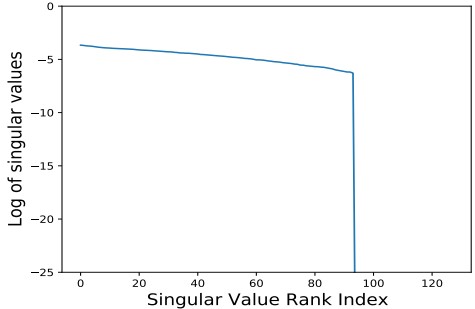

Figure 2: Singular value spectrum of the embedding space. The embedding vectors are computed from a pretrained SimCLR model on the validation set of ImageNet. Each embedding vector has a dimension of 128. The spectrum contains the singular values of the covariance matrix of these embedding vectors in sorted order and logarithmic scale. A number of singular values drop to zero, indicating collapsed dimensions.

## 4   DIMENSIONAL COLLAPSE CAUSED BY STRONG AUGMENTATION

### 4.1   LINEAR MODEL

In this section, we explain one scenario for contrastive learning to have collapsed embedding dimensions, where the augmentation surpasses the input information. We focus on a simple linear network setting. We denote the input vector as $\mathbf{x}$ and the augmentation is an additive noise. The network is a single linear layer with weight matrix is $W$. Hence, the embedding vector is $\mathbf{z} = W\mathbf{x}$. We focus on a typical contrastive loss, InfoNCE (van den Oord et al., 2018):

$$L = - \sum_{i=1}^{N} \log \frac{\exp(-|\mathbf{z}_i - \mathbf{z}_i'|^2/2)}{\sum_{j \neq i} \exp(-|\mathbf{z}_i - \mathbf{z}_j|^2/2) + \exp(-|\mathbf{z}_i - \mathbf{z}_i'|^2/2)} \quad (2)$$

where $\mathbf{z}_i$ and $\mathbf{z}_i'$ are a pair of embedding vectors from the two branches, $\mathbf{z}_j$ indicates the negative samples within the minibatch. When all $\mathbf{z}_i$ and $\mathbf{z}_i'$ are normalized to be unit vector, the negative distance $-|\mathbf{z}_i - \mathbf{z}_i'|^2/2$ can be replaced by inner products $\mathbf{z}_i^T \mathbf{z}_i'$. The model is trained with a basic stochastic gradient descent without momentum or weight decay.

## 4.2 Gradient Flow Dynamics

We study the dynamics via gradient flow, i.e., gradient descent with an infinitesimally small learning rate.

**Lemma 1.** *The weight matrix in a linear contrastive self-supervised learning model evolves by:*

$$\dot{W} = -G \tag{3}$$

*where $G = \sum_i (\boldsymbol{g}_{\boldsymbol{z}_i} \boldsymbol{x}_i^T + \boldsymbol{g}_{\boldsymbol{z}_i'} \boldsymbol{x}_i'^T)$, and $\boldsymbol{g}_{\boldsymbol{z}_i}$ is the gradient on the embedding vector $\boldsymbol{z}_i$ (similarly $\boldsymbol{g}_{\boldsymbol{z}_i'}$).*

This can be easily proven based on the chain rule. See proof in Appendix B.1. For InfoNCE loss defined in Eqn 2, the gradient of the embedding vector for each branch can be written as

$$\mathbf{g}_{\mathbf{z}_i} = \sum_{j \neq i} \alpha_{ij}(\mathbf{z}_j - \mathbf{z}_i') + \sum_{j \neq i} \alpha_{ji}(\mathbf{z}_j - \mathbf{z}_i), \qquad \mathbf{g}_{\mathbf{z}_i'} = \sum_{j \neq i} \alpha_{ij}(\mathbf{z}_i' - \mathbf{z}_i) \tag{4}$$

where $\{\alpha_{ij}\}$ are the softmax of similarity of between $\boldsymbol{z}_i$ and $\{\boldsymbol{z}_j\}$, defined by $\alpha_{ij} = \exp(-|\mathbf{z}_i - \mathbf{z}_j|^2/2)/Z_i$, $\alpha_{ii} = \exp(-|\mathbf{z}_i - \mathbf{z}_i'|^2/2)/Z_i$, and $Z_i = \sum_{j \neq i} \exp(-|\mathbf{z}_i - \mathbf{z}_j|^2/2) + \exp(-|\mathbf{z}_i - \mathbf{z}_i'|^2/2)$. Hence, $\sum_j \alpha_{ij} = 1$. Since $\boldsymbol{z}_i = W\mathbf{x}_i$, we have

$$G = -WX \tag{5}$$

where

$$X := -\sum_i \left( \sum_{j \neq i} \alpha_{ij}(\mathbf{x}_i' - \mathbf{x}_j) + \sum_{j \neq i} \alpha_{ji}(\mathbf{x}_i - \mathbf{x}_j) \right) \mathbf{x}_i^T - \sum_i (1 - \alpha_{ii})(\mathbf{x}_i' - \mathbf{x}_i)\mathbf{x}_i'^T \tag{6}$$

**Lemma 2.** *$X$ is a difference of two PSD matrices:*

$$X = \hat{\Sigma}_0 - \hat{\Sigma}_1 \tag{7}$$

*Here $\hat{\Sigma}_0 = \sum_{i,j} \alpha_{ij}(\boldsymbol{x}_i - \boldsymbol{x}_j)(\boldsymbol{x}_i - \boldsymbol{x}_j)^T$ is a weighted data distribution covariance matrix and $\hat{\Sigma}_1 = \sum_i (1 - \alpha_{ii})(\boldsymbol{x}_i' - \boldsymbol{x}_i)(\boldsymbol{x}_i' - \boldsymbol{x}_i)^T$ is a weighted augmentation distribution covariance matrix.*

See proof in Appendix B.2. Therefore, the amplitude of augmentation determines whether $X$ is a positive definite matrix. Similar to Theorem 3-4 in Tian et al. (2020), Lemma 2 also models the time derivative of weight $W$ as a product of $W$ and a symmetric and/or PSD matrices. However, Lemma 2 is much more general: it applies to InfoNCE with multiple negative contrastive terms, remains true when $\alpha_{ij}$ varies with sample pair $(i, j)$, and holds with finite batch size $N$. In contrast, Theorem 4 in Tian et al. (2020) only works for one negative term in InfoNCE, holds only in the population sense (i.e., $N \to +\infty$), and the formulation has residual terms, if $\alpha_{ij}$ are not constants.

Next, we look into the dynamics of weight matrix $W$ given property of $X$.

**Theorem 1.** *With fixed matrix $X$ (defined in Eqn 6) and strong augmentation such that $X$ has negative eigenvalues, the weight matrix $W$ has vanishing singular values.*

See proof in Appendix B.3.

**Corollary 1** (Dimensional Collapse Caused by Strong Augmentation). *With strong augmentation, the embedding space covariance matrix becomes low-rank.*

The embedding space is identified by the singular value spectrum of the covariance matrix on the embedding (Eqn. 1), $C = \sum_i (\mathbf{z}_i - \bar{\mathbf{z}})(\mathbf{z}_i - \bar{\mathbf{z}})^T/N = \sum_i W(\mathbf{x}_i - \bar{\mathbf{x}})(\mathbf{x}_i - \bar{\mathbf{x}})^T W^T/N$. Since $W$ has vanishing singular values, $C$ is also low-rank, indicating collapsed dimensions.

Numerical simulation verifies our theory. We choice input data as isotropic Gaussian with covariance matrix $\sum_{i,j} (\mathbf{x}_i - \mathbf{x}_j)(\mathbf{x}_i - \mathbf{x}_j)^T/N = I$. We set the augmentation as additive Gaussian with covariance matrix equal to $\sum_i (\mathbf{x}_i' - \mathbf{x}_i)(\mathbf{x}_i' - \mathbf{x}_i)^T/N = block\_diagonal(\mathbf{0}, k*I)$, where the block has the size of 8x8. We plot the weight matrix singular value spectrum in Figure 3 with various augmentation amplitude $k$. This proves that under linear network setting, strong augmentation leads to dimensional collapse in embedding space.

Our theory in this section is limited to linear network settings. For more complex nonlinear networks, the collapsing condition will still depend on "strong augmentation" but interpreted differently. A strong augmentation will be determined by more complicated properties of the augmentation (higher-order statistics of augmentation, manifold property of augmentation vs. data distribution) conditioned on the capacity of the networks.

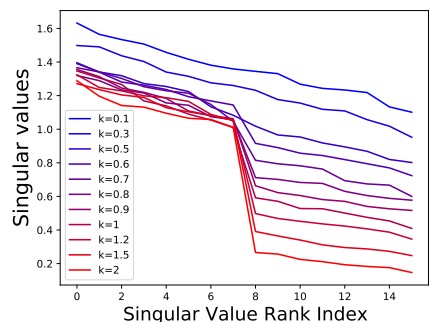

Figure 3: Weight matrix singular value spectrum with different augmentation amplitude $k$. The setting is a single layer linear toy model with each weight matrix of the size of 16x16, where the block has the size of 8x8. Strong augmentation results in vanishing singular values in weight matrices.

## 5 DIMENSIONAL COLLAPSE CAUSED BY IMPLICIT REGULARIZATION

### 5.1 TWO-LAYER LINEAR MODEL

With strong augmentation, a linear model under InfoNCE loss will have dimensional collapse. However, such scenarios rely on the condition that the network has a limited capacity which may not hold for real cases. On the other hand, when there is no strong augmentation ($\hat{\Sigma}_1 \prec \hat{\Sigma}_0$) and thus $X$ matrix remains PSD, a single linear model won't have dimensional collapsing. However, interestingly, for deep networks, dimensional collapsing still happens in practice. In the following, we will show that it stems from a different nature: implicit regularization, where over-parametrized linear networks tend to find low-rank solutions.

To understand this counter-intuitive phenomena, we start with the simplest **over-parametrized** setting by choosing the network as a two-layer linear MLP without bias. The weight matrices of these two layers are denoted by $W_1 \in \mathbb{R}^{d \times d}$ and $W_2 \in \mathbb{R}^{d \times d}$. Similar to the setting in Sec 4, the input vector is denoted as $\mathbf{x}$ and the augmentation is an additive noise. The embedding vector from each branch is $\mathbf{z} = W_2 W_1 \mathbf{x}$, hence $\mathbf{z} \in \mathbb{R}^n$. We do not normalize $\mathbf{z}$. See Figure 4. We use InfoNCE loss defined in Eqn 2. The model is trained with a basic stochastic gradient descent without momentum or weight decay.

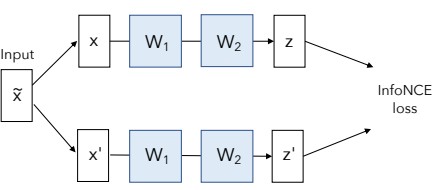

Figure 4: Two-layer Linear Model

### 5.2 GRADIENT FLOW DYNAMICS

Similar to Lemma 1, we derive the gradient flow on the two weight matrices $W_1$ and $W_2$.

**Lemma 3.** *The weight matrices of the two layer linear contrastive self-supervised learning model evolves by ($G = \sum_i (\mathbf{g}_{\mathbf{z}_i} \mathbf{x}_i^T + \mathbf{g}_{\mathbf{z}_i'} \mathbf{x}_i'^T)$ is defined in Lemma 1):*

$$\dot{W}_1 = -W_2^T G, \qquad \dot{W}_2 = -G W_1^T \tag{8}$$

This can be easily proven based on the chain rule. See proof in Appendix B.4. For the two layer case, similar to Eqn 5, we have the specific form of $G$:

$$G = -W_2 W_1 X \tag{9}$$

where $X$ is defined in Eqn 6. According to Lemma 2, we know that with small augmentation, $X = \hat{\Sigma}_0 - \hat{\Sigma}_1 \succ 0$ is a positive-definite matrix.

### 5.3 WEIGHT ALIGNMENT

Since we have two matrices $W_1$ and $W_2$, the first question is how they interact with each other. We apply singular value decomposition on both matrices $W_1$ and $W_2$, i.e., $W_1 = U_1 S_1 V_1^T$, $W_2 = U_2 S_2 V_2^T$ and $S_1 = diag([\sigma_1^k])$, $S_2 = diag([\sigma_2^k])$. The alignment is now governed by the interaction

between the adjacent orthonormal matrices $V_2 := [\mathbf{v}_2^k]$ and $U_1 = [\mathbf{u}_1^k]$. This can be characterized by the *alignment matrix* $A = V_2^T U_1$, whose $(k, k')$-entry represents the alignment between the $k$-th right singular vector $\mathbf{v}_2^k$ of $W_2$ and the $k'$-th left singular vector $\mathbf{u}_1^{k'}$ of $W_1$. The following shows that indeed $W_1$ and $W_2$ aligns.

**Theorem 2** (Weight matrices align). *If for all $t$, $W_2(t)W_1(t) \neq 0$, $X(t)$ is positive-definite and $W_1(+\infty)$, $W_2(+\infty)$ have distinctive singular values, then the alignment matrix $A = V_2^T U_1 \to I$.*

See proof in Appendix B.5. Here, we also empirically demonstrate that under InfoNCE loss, the absolute value of the alignment matrix $A$ converges to an identity matrix. See Figure 5.

The alignment effect has been studied in other scenarios (Ji & Telgarsky, 2019; Radhakrishnan et al., 2020). In the real case, when some of our assumptions are not satisfied, e.g., there are degenerate singular values in weight matrices, we will not observe a perfect alignment. This can be easily understood by the fact that the singular decomposition is no longer unique given degenerate singular values. In our toy experiment, we specifically initialize the weight matrices to have non-degenerate singular values. In real scenario, when weight matrices are randomly initialized, we will only observe the alignment matrix to converge to a block-diagonal matrix, with each block representing a group of degenerate singular values.

Given the fact that singular vectors corresponding to the same singular value align, we can now study the dynamics of the singular values of each weight matrix $W_1$ and $W_2$.

**Theorem 3.** *If $W_2$ and $W_1$ are aligned (i.e., $V_2 = U_1^T$), then the singular values of the weight matrices $W_1$ and $W_2$ under InfoNCE loss evolve by:*

$$\dot{\sigma}_1^k = \sigma_1^k (\sigma_2^k)^2 (\mathbf{v}_1^{k^T} X \mathbf{v}_1^k) \qquad (10)$$

$$\dot{\sigma}_2^k = \sigma_2^k (\sigma_1^k)^2 (\mathbf{v}_1^{k^T} X \mathbf{v}_1^k) \qquad (11)$$

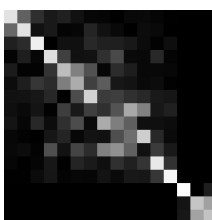

Figure 5: Visualization of the alignment matrix $A = V_2^T U_1$ after training. The setting is a 2-layer linear toy model with each weight matrix of the size of 16x16. The alignment matrix converges to an identity matrix.

See proof in Appendix B.6. According to Eqn. 10, $(\sigma_1^k)^2 = (\sigma_2^k)^2 + C$. We solve the singular value dynamics analytically: $\dot{\sigma}_1^k = \sigma_1^k((\sigma_1^k)^2 + C)(\mathbf{v}_1^{k^T} X \mathbf{v}_1^k)$. This shows that a pair of singular values (singular values with same ranking from the other matrix) have gradients proportional to themselves. Notice that $X$ is a positive definite matrix, the term $\mathbf{v}_1^{k^T} X \mathbf{v}_1^k$ is always non-negative. This explains why we observe that the smallest group of singular values grow significantly slower. See demonstrative experiment results in Figure 6a and 6b.

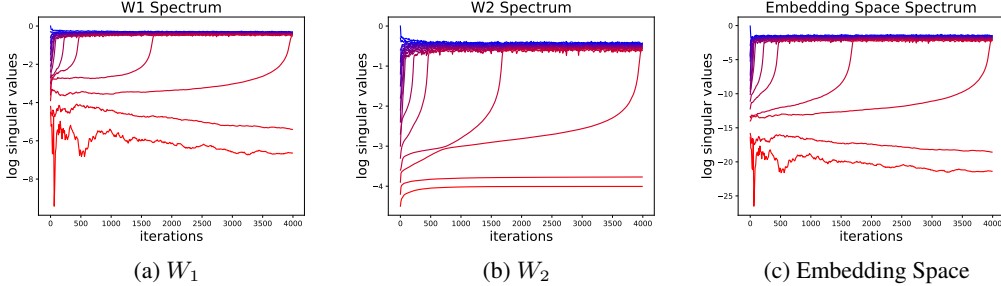

(a) $W_1$         (b) $W_2$         (c) Embedding Space

Figure 6: Evolution of the singular values of the weight matrices and the embedding space covariance matrix. The setting is a 2-layer linear toy model with each weight matrix of the size of 16x16. The lowest few singular values of each weight matrix remain significantly smaller.

**Corollary 2** (Dimensional Collapse Caused by Implicit Regularization). *With small augmentation and over-parametrized linear networks, the embedding space covariance matrix becomes low-rank.*

The embedding space is identified by the singular value spectrum of the covariance matrix on the embedding vectors, $C = \sum (\mathbf{z} - \bar{\mathbf{z}})(\mathbf{z} - \bar{\mathbf{z}})^T / N = \sum W_2 W_1 (\mathbf{x} - \bar{\mathbf{x}})(\mathbf{x} - \bar{\mathbf{x}})^T W_1^T W_2^T / N$. As

$W_2 W_1$ evolves to be low-rank, $C$ is low-rank, indicating collapsed dimensions. See Figure 6c for experimental verification.

Our theory can also be extended to multilayer networks and nonlinear setting. Please see Appendix C

## 6   DIRECTCLR

### 6.1   MOTIVATION

We now leverage our theoretical finding to design novel algorithms. Here we are targeting the projector component in contrastive learning.

Empirically, adding a projector substantially improves the quality of the learned representation and downstream performance (Chen et al., 2020a). Checking the spectrum of the representation layer also reveals a difference with/without a projector. To see this, we train two SimCLR models with and without a projector. The representation space spectrum are shown in Figure 7b. The dimensional collapse in representation space happens when the model is trained without a projector. Thus, the projector prevents the collapse in the representation space.

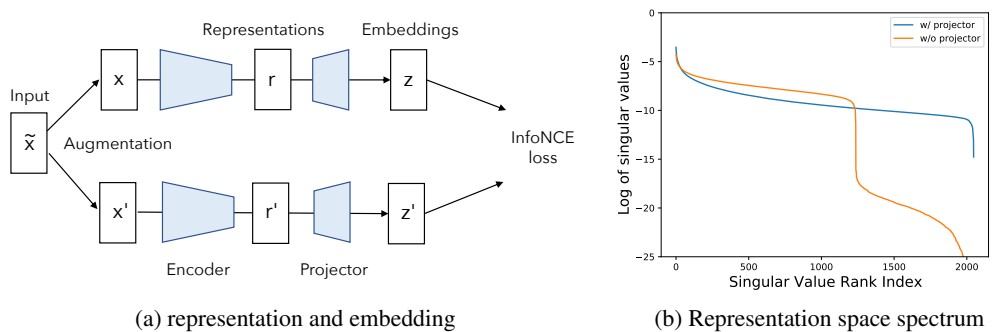

(a) representation and embedding

(b) Representation space spectrum

Figure 7: (a) Definition of representation and the embedding space; (b) Singular value spectrums of the representation space of pretrained contrastive learning models (pretrained with or without a projector). The representation vectors are the output from the ResNet50 encoder and directly used for downstream tasks. Each representation vector has a dimension of 2048. Without a projector, SimCLR suffers from dimensional collapse in the representation space.

The projector in contrastive learning is essential to prevent dimensional collapse in the representation space. We claim the following propositions regarding a linear projector in contrastive learning models.

**Proposition 1.** *A linear projector weight matrix only needs to be **diagonal**.*

**Proposition 2.** *A linear projector weight matrix only needs to be **low-rank**.*

Based on our theory on implicit regularization dynamics, we expect to see adjacent layers $W_1 (= U_1 S_1 V_1^T)$ and $W_2 (= U_2 S_2 V_2^T)$ to be aligned such that the overall dynamics is only governed by their singular values $S_1$ and $S_2$. And the orthogonal matrices $V_2^T$ and $U_1$ are redundant as they will evolve to $V_2^T U_1 = I$, given $S_1$ and $S_2$.

Now, let's consider the linear projector SimCLR model and only focus on the channel dimension. $W_1$ is the last layer in the encoder, and $W_2$ is the projector weight matrix. Our propositions claim that for this projector matrix $W_2$, the orthogonal component $V_2$ can be omitted. Because the previous layer $W_1$ is fully trainable, its orthogonal component ($U_1$) will always evolve to satisfy $V_2^T U_1 = I$. Therefore, the final behavior of the projector is only determined by the singular values ($S_2$) of the projector weight matrix. This motivates Proposition 1: the orthogonal component of the weight matrix doesn't matter. So we can set the projector matrix as a diagonal matrix.

Also, according to our theory, the weight matrix will always converge to the low-rank. The singular value diagonal matrix naturally becomes low-rank, so why not just set it low-rank directly? This is the motivation of Proposition 2.

These propositions are verified via ablation studies in Sec 6.3. Given these two propositions, we propose *DirectCLR*, which is effectively using a low-rank diagonal projector.

## 6.2 MAIN IDEA

We propose to remove the projector in contrastive learning by directly sending a sub-vector of the representation vector to the loss function. We call our method *DirectCLR*. In contrast to all recent state-of-the-art self-supervised learning methods, our method directly optimizes the representation space. See Figure 8, *DirectCLR* picks a subvector of the representation $\mathbf{z} = \mathbf{r}[0 : d_0]$, where $d_0$ is a hyperparameter. Then, it applies a standard InfoNCE loss on this normalized subvector $\hat{\mathbf{z}} = \mathbf{z}/|\mathbf{z}|$, $L = \sum_i \log \frac{\exp(\hat{\mathbf{z}}_i \cdot \hat{\mathbf{z}}'_i)}{\sum_j \exp(\hat{\mathbf{z}}_i \cdot \hat{\mathbf{z}}_j)}$.

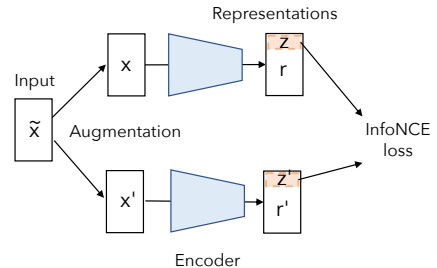

Figure 8: *DirectCLR*: no trainable projector, simply apply InfoNCE loss on the a fixed sub-vector of the representations

We train *DirectCLR* with a standard recipe of Sim-CLR for 100 epochs on ImageNet. The backbone encoder is a ResNet50. More implementation details can be found in the Appendix D. *DirectCLR* demonstrates better performance compared to SimCLR with a trainable linear projector on ImageNet. The linear probe accuracies for each model are listed in Table 1.

| Loss function | Projector | Accuracy |
|---|---|---|
| SimCLR | 2-layer nonlinear projector | 66.5 |
| SimCLR | 1-layer linear projector | 61.1 |
| SimCLR | no projector | 51.5 |
| *DirectCLR* | no projector | 62.7 |

Table 1: Linear probe accuracy on ImageNet. Each model is trained on ImageNet for 100 epochs with standard training recipe. The backbone encoder is a ResNet50. *DirectCLR* outperforms SimCLR with 1-layer linear projector.

We visualize the learnt representation space spectrum in Figure 9. *DirectCLR* prevents dimensional collapse in the representation space similar to the functionality of a trainable projector in SimCLR.

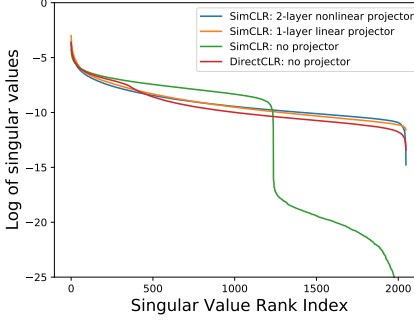

Figure 9: Representation space spectrum of *DirectCLR* compared to SimCLR (a) with a 2-layer nonlinear projector (b) with a 1-layer linear projector (c) without projector. The spectrums are computed based on the output from the backbone, using ImgaeNet validation set. Similar to Sim-CLR with projectors, *DirectCLR* is able to prevent dimensional collapse in the representation space.

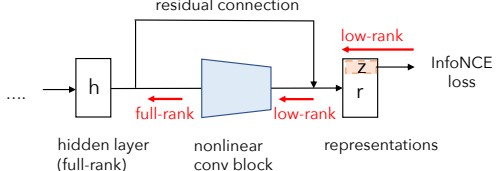

Figure 10: Why is the whole representation vector $\mathbf{r}$ meaningful in *DirectCLR* while only part of it receives gradient? It takes advantage of the residual connection in the backbone. Thus, the gradient passing through the representation vector is low-rank where only the first $d_0$ channel dimensions are non-zero. When the gradient enters the ResNet backbone and passes through the last nonlinear conv block, it becomes full rank. Therefore, this hidden layer $\mathbf{h}$ receives gradients on all channels. During forward pass, $\mathbf{h}$ is directly fed to the representation vectors via the residual connection. Therefore, the entire representation vector $\mathbf{r}$ is meaningful.

One may suspect that the contrastive loss in *DirectCLR* does not apply a gradient on the rest part of the representation vector $\mathbf{r}[d_0 :]$, then why these dimensions would contain useful information?

Here, we show that the entire representation vector $\mathbf{r}$ contains useful information. See Figure 10. First, the gradient backpropagating through the representation vector is low-rank, where only the first $d_0$ channel dimensions are non-zero. When the gradient enters the ResNet backbone and passes through the last nonlinear conv block, it becomes full rank. Therefore, this hidden layer $\mathbf{h}$ receives gradients on all channels. Note that $\mathbf{h}$ and $\mathbf{r}$ have a same channel dimension of 2048. Next, we consider the forward pass. This hidden layer $\mathbf{h}$ is directly fed to the representation vectors via the residual connection. As a result, the rest part of the representation vector $\mathbf{r}[d_0 :]$ is not trivial. In addition, we run an ablation study in Sec F to test the linear probe accuracy based only on the "directly" optimized vector. This verifies that the whole representation vector is meaningful.

## 6.3 ABLATION STUDY

| Projector | diagonal | low-rank | Top-1 Accuracy |
|---|---|---|---|
| no projector | | | 51.5 |
| orthogonal projector | | | 52.2 |
| trainable projector | | | 61.1 |
| trainable diagonal projector | ✓ | | 60.2 |
| fixed low-rank projector | | ✓ | 62.3 |
| fixed low-rank diagonal projector | ✓ | ✓ | 62.7 |

Table 2: Ablation study: top-1 accuracies on ImageNet by SimCLR model with different projector settings.

To further verify our hypothesis, we have perform ablation studies.

Proposition 1 matches the fact that: (a) an orthogonal constrained projector performs the same as the non-projector setting; (b) fixed low-rank projector performs the same as a fixed diagonal projector; (c) trainable linear projector performs the same as a trainable diagonal projector.

Proposition 2 matches the observation that a low-rank projector has the highest accuracy.

Please see more detailed ablation study discuss and additional ablation experiments in Appendix F.

## 7 CONCLUSIONS

In this work, we showed that contrastive self-supervised learning suffers from dimensional collapse, where the embedding vectors only span a lower-dimensional subspace. We provided the theoretical understanding of this phenomenon and showed that there are two mechanisms causing dimensional collapse: strong augmentation and implicit regularization. Inspired by our theory, we proposed a novel contrastive self-supervised learning method *DirectCLR* that directly optimizes the representation space without relying on a trainable projector. *DirectCLR* outperforms SimCLR with a linear projector on ImageNet.

## ACKNOWLEDGEMENT

We thank Yubei Chen, Jiachen Zhu, Adrien Bardes, Nicolas Ballas, Randall Balestriero, Quentin Garrido for useful discussions.

## REPRODUCIBILITY STATEMENT

We provide detailed proof for all the lemmas and theorems in the Appendices. Code (in PyTorch) is available at https://github.com/facebookresearch/directclr

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

## A    USEFUL LEMMAS

We adapt two useful lemmas from Arora et al. (2019a).

**Lemma 4.** *Given a matrix $W$ and the dynamics that $W$ evolves by $\dot{W}$, the singular values of this matrix evolve by:*

$$\dot{\sigma^k} = \boldsymbol{u}^{k^T} \dot{W} \boldsymbol{v}^k \tag{12}$$

*where $\boldsymbol{u}^k$ and $\boldsymbol{v}^k$ are singular value $\sigma^k$'s corresponding left and right singular vectors. i.e. the $k$-th column of matrices $U$ and $V$ respectively.*

*Proof.* Given a matrix $W$ and its singular value decomposition $W = USV^T$. We have the dynamics of the matrix

$$\dot{W} = \dot{U}SV^T + U\dot{S}V^T + US\dot{V}^T$$

Multiplying $U^T$ from the left and multiplying $V$ from the right, considering $U$ and $V$ are orthogonal matrices, we have

$$U^T\dot{W}V = U^T\dot{U}S + \dot{S} + S\dot{V}^TV$$

Since $S = diag(\sigma^k)$ is a diagonal matrix, we have

$$\dot{\sigma^k} = \mathbf{u}^{k^T}\dot{W}\mathbf{v}^k - \mathbf{u}^{k^T}\dot{\mathbf{u}}^k\sigma^k - \sigma^k\dot{\mathbf{v}}^{k^T}\mathbf{v}^k$$

Again, considering $\mathbf{u}^k$ and $\mathbf{v}^k$ have unit-norm, we have $\mathbf{u}^{k^T}\dot{\mathbf{u}}^k = 0$ and $\dot{\mathbf{v}}^{k^T}\mathbf{v}^k = 0$. Therefore, we derive

$$\dot{\sigma^k} = \mathbf{u}^{k^T}\dot{W}\mathbf{v}^k$$

$\square$

**Lemma 5.** *Given a matrix $W$ and the dynamics that $W$ evolves by $\dot{W}$, the singular vectors of this matrix evolve by:*

$$\dot{U} = U(H \odot (U^T\dot{W}VS + SV^T\dot{W}^TU)) \tag{13}$$
$$\dot{V} = V(H \odot (V^T\dot{W}^TUS + SU^T\dot{W}V)) \tag{14}$$

*where $\odot$ represents Hadamard element-wise multiplication. $H$ is a skew-symmetric matrix*

$$H^{k,k'} = \begin{cases} 1/(\sigma^{k^2} - \sigma^{k'^2}) & \text{if } k \neq k' \\ 0 & \text{if } k = k' \end{cases} \tag{15}$$

*Proof.* Same as proof for Lemma 1, we start from the following equation

$$U^T\dot{W}V = U^T\dot{U}S + \dot{S} + S\dot{V}^TV$$

Considering the fact that $U^T\dot{U}$ and $\dot{V}^TV$ are skew-symmetric matrices, whose diagonal terms are all zero, we Hadamard-multiply $\bar{I}$ to both sides of the equation. Here, $\bar{I}$ has all diagonal values equal zeros and all off-diagonal values equal to one, we have

$$\bar{I} \odot U^T\dot{W}V = U^T\dot{U}S + S\dot{V}^TV \tag{16}$$

Taking transpose, we have

$$\bar{I} \odot V^T\dot{W}U = -SU^T\dot{U} - \dot{V}^TVS \tag{17}$$

Right-multiplying $S$ to Eqn 16 and left-multiplying $S$ to Eqn 17, then adding them up, we have

$$U^T\dot{U}S^2 - S^2U^T\dot{U} = \bar{I} \odot (U^T\dot{W}VS + SV^T\dot{W}U)$$

Therefore, we have

$$\dot{U} = U(H \odot (U^T\dot{W}VS + SV^T\dot{W}^TU))$$

where

$$H^{k,k'} = \begin{cases} 1/(\sigma^{k^2} - \sigma^{k'^2}) & \text{if } k \neq k' \\ 0 & \text{if } k = k' \end{cases}$$

Similar proof applies to Eqn 14. $\square$

**Lemma 6** (Alignment matrix dynamics). *The alignment matrix $A$, defined by $A = V_2^TU_1$, evolves by:*

$$\dot{A} = -A(H_1 \odot (A^TF + F^TA)) + (H_2 \odot (AF^T + FA^T))A \tag{18}$$

*where $\odot$ represents Hadamard (element-wise) multiplication. $H_l$ is a skew-symmetric matrix, whose $(k, k')$-entry is given by*

$$H_l^{k,k'} = \begin{cases} 1/(\sigma_l^{k^2} - \sigma_l^{k'^2}) & \text{if } k \neq k' \\ 0 & \text{if } k = k' \end{cases} \tag{19}$$

*and $F$ is defined by*

$$F = S_2U_2^TGV_1S_1 \tag{20}$$

*Proof.* According to Lemma. 5, we have

$$\dot{U}_1 = U_1(H_1 \odot (U_1^T \dot{W}_1 V_1 S_1 + S_1 V_1^T \dot{W}_1^T U_1))$$
$$\dot{V}_2 = V_2(H_2 \odot (V_2^T \dot{W}_2^T U_2 S_2 + S_2 U_2^T \dot{W}_2 V_2))$$

Plugging the above two equations and Eqn 8, the dynamics of the alignment matrix $A = V_2^T U_1$ can be written as

$$
\begin{aligned}
\dot{A} &= V_2^T \dot{U}_1 + \dot{V}_2^T U_1 \\
&= V_2^T U_1(H_1 \odot (U_1^T \dot{W}_1 V_1 S_1 + S_1 V_1^T \dot{W}_1^T U_1)) + (H_2 \odot (V_2^T \dot{W}_2^T U_2 S_2 + S_2 U_2^T \dot{W}_2 V_2))^T V_2^T U_1 \\
&= -A(H_1 \odot (U_1^T W_2^T G V_1 S_1 + S_1 V_1^T G^T W_2 U_1)) + (H_2 \odot (S_2 U_2^T G W_1^T V_2 + V_2^T W_1 G^T U_2 S_2))A \\
&= -A(H_1 \odot (U_1^T V_2 S_2 U_2^T G V_1 S_1 + S_1 V_1^T G^T U_2 S_2 V_2^T U_1)) \\
&\quad + (H_2 \odot (S_2 U_2^T G V_1 S_1 U_1^T V_2 + V_2^T U_1 S_1 V_1^T G^T U_2 S_2))A \\
&= -A(H_1 \odot (A^T S_2 U_2^T G V_1 S_1 + S_1 V_1^T G^T U_2 S_2 A) \\
&\quad + (H_2 \odot (S_2 U_2^T G V_1 S_1 A^T + A S_1 V_1^T G^T U_2 S_2))A \\
&= -A(H_1 \odot (A^T F + F^T A)) + (H_2 \odot (A F^T + F A^T))A
\end{aligned}
$$

where

$$F = S_2 U_2^T G V_1 S_1$$

$\square$

**Lemma 7** (Singular value dynamics). *The singular values of the weight matrices $W_1$ and $W_2$ evolve by:*

$$\dot{\sigma}_1^k = -\sum_{k'} (\boldsymbol{v}_2^{k'T} \boldsymbol{u}_1^k) \sigma_2^{k'} (\boldsymbol{u}_2^{k'T} G \boldsymbol{v}_1^k) \tag{21}$$

$$\dot{\sigma}_2^k = -\sum_{k'} (\boldsymbol{u}_1^{k'T} \boldsymbol{v}_2^k) \sigma_1^{k'} (\boldsymbol{u}_2^{kT} G \boldsymbol{v}_1^{k'}) \tag{22}$$

*Proof.* According to Lemma 4,

$$\dot{\sigma}_1^r = \boldsymbol{u}_1^{rT} \dot{W}_1 \boldsymbol{v}_1^r$$

Plugging in Eqn 8, we have

$$
\begin{aligned}
\dot{\sigma}_1^k &= -\mathbf{u}_1^{kT} W_2^T G \mathbf{v}_1^k \\
&= -\mathbf{u}_1^{kT} V_2 S_2 U_2^T G \mathbf{v}_1^k \\
&= -\sum_{k'} (\mathbf{v}_2^{k'T} \mathbf{u}_1^k) \sigma_2^{k'} (\mathbf{u}_2^{k'T} G \mathbf{v}_1^k)
\end{aligned}
$$

Similar proof applies to Eqn 22. $\square$

## B  DELAYED PROOFS

### B.1  PROOF OF LEMMA 1

The gradient on matrix $W$ is

$$\frac{dL}{dW} = \sum_i \left( \frac{\partial L}{\partial \mathbf{z}_i} \frac{\partial \mathbf{z}_i}{\partial W} + \frac{\partial L}{\partial \mathbf{z}_i'} \frac{\partial \mathbf{z}_i'}{\partial W} \right)$$

We denote the gradient on $\mathbf{z}_i$ and $\mathbf{z}_i'$ as $\mathbf{g}_{\mathbf{z}_i}$ and $\mathbf{g}_{\mathbf{z}_i'}$, respectively. Since $\frac{\partial \mathbf{z}_i}{\partial W} = \mathbf{x}_i$ and $\frac{\partial \mathbf{z}_i'}{\partial W} = \mathbf{x}_i'$, we get

$$\dot{W} = -\left(\frac{dL}{dW}\right)^T = -\sum_i (\mathbf{g}_{\mathbf{z}_i} \mathbf{x}_i^T + \mathbf{g}_{\mathbf{z}_i'} \mathbf{x}_i'^T)$$

## B.2 PROOF OF LEMMA 2

*Proof.* $X$ is defined in Eqn 6.

$$
\begin{aligned}
X &= \sum_i (\sum_{j\neq i} \alpha_{ij}(\mathbf{x}'_i - \mathbf{x}_j) + \sum_{j\neq i} \alpha_{ji}(\mathbf{x}_i - \mathbf{x}_j))\mathbf{x}_i^T - \sum_i (1-\alpha_{ii})(\mathbf{x}'_i - \mathbf{x}_i)\mathbf{x}'^T_i \\
&= \sum_i \sum_{j\neq i} \alpha_{ij}\mathbf{x}'_i\mathbf{x}_i^T - \sum_i \sum_{j\neq i} \alpha_{ij}\mathbf{x}_j\mathbf{x}_i^T + \sum_i \sum_{j\neq i} \alpha_{ji}(\mathbf{x}_i - \mathbf{x}_j)(\mathbf{x}_i - \mathbf{x}_j)^T \\
&\quad + \sum_i \sum_{j\neq i} \alpha_{ji}(\mathbf{x}_i - \mathbf{x}_j)\mathbf{x}_j^T - \sum_i (1-\alpha_{ii})(\mathbf{x}'_i - \mathbf{x}_i)(\mathbf{x}'_i - \mathbf{x}_i)^T - \sum_i (1-\alpha_{ii})(\mathbf{x}'_i - \mathbf{x}_i)\mathbf{x}_i^T
\end{aligned}
$$

Given the fact that $\sum_{j\neq i} \alpha_{ij} = 1 - \alpha_{ii}$, we have $\sum_i \sum_{j\neq i} \alpha_{ij}\mathbf{x}'_i\mathbf{x}_i^T = \sum_i (1-\alpha_{ii})\mathbf{x}'_i\mathbf{x}_i^T$. Also, since $\sum_i \sum_{j\neq i}$ iterates all pairs of $i,j$, we can replace the index between $i$ and $j$, we have $\sum_i \sum_{j\neq i} \alpha_{ij}\mathbf{x}_j\mathbf{x}_i^T = \sum_i \sum_{j\neq i} \alpha_{ji}\mathbf{x}_i\mathbf{x}_j^T$.

Therefore

$$
X = \sum_i \sum_{j\neq i} \alpha_{ji}(\mathbf{x}_i - \mathbf{x}_j)(\mathbf{x}_i - \mathbf{x}_j)^T - \sum_i (1-\alpha_{ii})(\mathbf{x}'_i - \mathbf{x}_i)(\mathbf{x}'_i - \mathbf{x}_i)^T
$$

$\square$

## B.3 PROOF OF THEOREM 1

*Proof.* According to Lemma 1, we have

$$
\frac{d}{dt}W = WX \tag{23}
$$

For a fixed $X$, we solve this equation analyically,

$$
W(t) = W(0)\exp(Xt)
$$

Apply eigen-decomposition on $X$, $X = U\Lambda U^T$. Then we have $\exp(Xt) = U\exp(\Lambda t)U^T$. Therefore,

$$
W(t) = W(0)U\exp(\Lambda t)U^T
$$

Because $X$ has negative eigenvalues, i.e., $\Lambda$ has negative terms, we have for $t \to \infty$, $\exp(\Lambda t)$ is rank deficient. Therefore, we know that $W(\infty)$ is also rank deficient, the weight matrix $W$ has vanishing singular values.

$\square$

## B.4 PROOF OF LEMMA 3

*Proof.* The gradient on matrix $W_2$ is

$$
\frac{dL}{dW_2} = \sum_i \left(\frac{\partial L}{\partial \mathbf{z}_i}\frac{\partial \mathbf{z}_i}{\partial W_2} + \frac{\partial L}{\partial \mathbf{z}'_i}\frac{\partial \mathbf{z}'_i}{\partial W_2}\right) \tag{24}
$$

We denote the gradient on $\mathbf{z}_i$ and $\mathbf{z}'_i$ as $\mathbf{g}_{\mathbf{z}_i}$ and $\mathbf{g}_{\mathbf{z}'_i}$, respectively. Since $\frac{\partial \mathbf{z}_i}{\partial W_2} = W_1\mathbf{x}_i$ and $\frac{\partial \mathbf{z}'_i}{\partial W_2} = W_1\mathbf{x}'_i$, we get

$$
\dot{W}_2 = -\left(\frac{dL}{dW_2}\right)^T = -\sum_i (\mathbf{g}_{\mathbf{z}_i}\mathbf{x}_i^T + \mathbf{g}_{\mathbf{z}'_i}\mathbf{x}'^T_i)W_1^T \tag{25}
$$

Similar proof applies to $W_1$.

$\square$

### B.5 PROOF OF THEOREM 2

Here, we prove that under the assumption that singular values are non-degenerate, the alignment matrix $A = V_2^T U_1$ converges to identity matrix.

*Proof.* According to Lemma 3, we have

$$\frac{d}{dt}(W_1 W_1^T) = -W_1 G^T W_2 - W_2^T G W_1^T$$

$$\frac{d}{dt}(W_2^T W_2) = -W_2^T G W_1^T - W_1 G^T W_2$$

therefore,

$$\frac{d}{dt}(W_1 W_1^T - W_2^T W_2) = 0$$

or

$$W_1 W_1^T - W_2^T W_2 = C$$

Next, we show that the Frobenius norm of each weight matrix grow to infinitely.

$$\frac{d}{dt}||W_1||_F^2 = \frac{d}{dt}tr(W_1 W_1^T) = -tr(W_2^T G W_1^T) - tr(W_1 G_1^T W_2)$$

According to Eqn 9, $G = -W_2 W_1 X$, we have

$$-tr(W_2^T G W_1^T) = tr(W_2^T W_2 W_1 X W_1^T)$$
$$= tr(W_2 W_1 X W_1^T W_2^T)$$

Because $X$ is a positive definite matrix and for all $t$, $W_2(t)W_1(t) \neq 0$, we know $B := W_2 W_1 X W_1^T W_2^T$ is positive semi-definite and $B \neq 0$. Therefore, $tr(B) = \sum_k \lambda_k(B) > 0$ since not all eigenvalues of $B$ are zero.

Therefore, we know $||W_1||_F^2 \to +\infty$ (similarly $||W_2||_F^2 \to +\infty$). In the limit $t->+\infty$, we have

$$W_1 W_1^T = W_2^T W_2$$

Plug in the singular value decomposition of $W_1$ and $W_2$, we have $U_1 S_1^2 U_1^T = V_2 S_2^2 V_2^T$. Assuming $W_1$ and $W_2$ have non-degenerate singular values, due to the uniqueness of eigen-decomposition, we have

$$U_1 = V_2$$

therefore,

$$V_2^T U_1 = I$$

$\square$

**Remark**. Note that when the non-degenerate singular value assumption does not hold, the corresponding singular vectors are not unique and we will not observe the corresponding dimensions becoming aligned.

### B.6 PROOF OF THEOREM 3

*Proof.* According to Theorem 2, for $\sigma_1^k$ and $\sigma_2^k$ with same index, the corresponding singular vector pairs $\mathbf{v}_2^k$ and $\mathbf{u}_1^k$ will get aligned, i.e., $\mathbf{v}_2^{k'^T} \mathbf{u}_1^k \to \delta_{i,j}$. Therefore, Eqn 21 and Eqn 22 can be simplified to

$$\dot{\sigma}_1^k \to -\sigma_2^k(\mathbf{u}_2^{k^T} G \mathbf{v}_1^k)$$
$$\dot{\sigma}_2^k \to -\sigma_1^k(\mathbf{u}_2^{k^T} G \mathbf{v}_1^k)$$

Insert Eqn 9 and considering the alignment, we derive

$$\dot{\sigma}_1^k \to \sigma_1^k(\sigma_2^k)^2(\mathbf{v}_1^{k^T} X \mathbf{v}_1^k)$$
$$\dot{\sigma}_2^k \to \sigma_2^k(\sigma_1^k)^2(\mathbf{v}_1^{k^T} X \mathbf{v}_1^k)$$

$\square$

## C  EFFECT OF MORE LAYERS AND NONLINEARITY

In our toy model, we focused on a two-layer linear MLP setting. Here, we empirically show that our theory extends to multilayer and nonlinear cases, as shown in Figure 11a.

Stronger over-parametrization leads to a stronger collapsing effect, which has been shown theoretically (Arora et al., 2019a; Barrett & Dherin, 2021) and empirically (Jing et al., 2020). This can be explained by the fact that more adjacent matrices getting aligned, and the collapsing in the product matrix gets amplified. Note that for a single-layer case, $L = 1$, there is no dimensional collapse in the embedding space, which is consistent with our analysis.

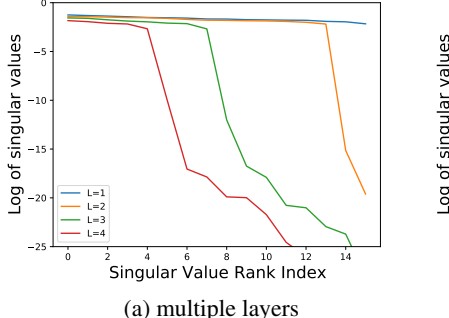 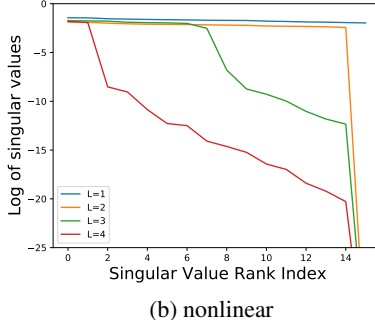

(a) multiple layers          (b) nonlinear

Figure 11: Embedding space singular value spectrum with different layers on (a) linear and (b) nonlinear networks. All models use weight matrices with a size of 16x16. Adding more layers in the network leads to more collapsed dimensions. Adding nonlinearity leads to a similar collapsing effect.

We empirically show that the collapsing effect also applies to the nonlinear scenario. We insert ReLU between linear layers and observe a similar singular value collapse compared to the linear case. See Figure 11b.

## D  IMPLEMENTATION DETAIL

### D.1  AUGMENTATIONS

Each input image is transformed twice to produce the two distorted views for contrastive loss. The image augmentation pipeline includes random cropping, resizing to 224x224, random horizontal flipping, color jittering, grayscale, Gaussian blurring, and solarization.

### D.2  NETWORK

Throughout the ImageNet experiments in this paper, we use a ResNet-50 (He et al., 2016) as an encoder. This network has an output of dimension 2048, which is called a representation vector.

### D.3  OPTIMIZATION

We use a LARS optimizer and train all models for 100 epochs. The batch size is 4096, which fits into 32 GPUs during training. The learning rate is 4.8 as in SimCLR (Chen et al., 2020a), which goes through a 10 epoch of warming up and then a cosine decay schedule.

## E  HYPERPARAMETER TUNING ON $d_0$

Here, we list the ImageNet accuracy with various $d_0$ value in Figure 12. It's easy to see that when $d_0 \to 0$, there's too little gradient information coming from the loss, the performance drops. When $d_0 \to 2048$, the model converges to standard SimCLR without a projector, which we know suffers from dimensional collapse in representation space.

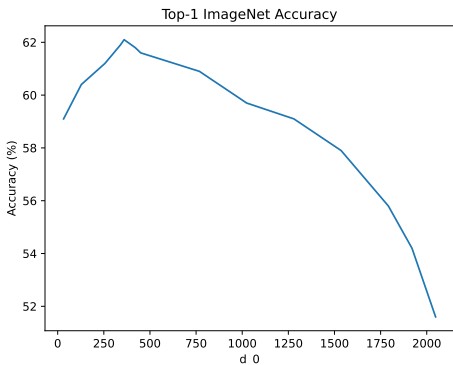

Figure 12: Hyperparameter tuning on $d_0$ based on ImageNet linear probe Top-1 accuracy.

## F  ABLATION STUDY DETAIL

**Fixed low-rank projector vs Fixed low-rank diagonal projector**: *DirectCLR* is equivalent to SimCLR with a fixed low-rank diagoanl projector. It performs the same as a SimCLR with fixed low-rank projector, which achieves $62.3\%$ linear probe accuracy. Specifically, the singular values of this low-rank matrix are set to have $d_0$ numbers of 1 and 0 for the rest, then left- and right- multiply a fixed orthogonal matrix. Therefore, their only difference is that this fixed projector has an extra fixed orthogonal matrix in between.

**Trainable projector vs trainable diagonal projector**: We trained a SimCLR model with a trainable projector that is constrained be diagonal. The model achieves $60.2\%$ linear probe accuracy on ImageNet, which is close to a SimCLR with a 1-layer linear projector.

**Orthogonal projector vs no projector**: We train a single layer projector SimCLR model with orthogonal constraint using ExpM parametrization (Casado & Martínez-Rubio, 2019). Therefore, the projector weight matrix has all singular values fixed to be 1. This model reaches $52.2\%$ accuracy on ImageNet which is close to a SimCLR without projector.

These ablation studies verify the propostion 1 that the SimCLR projector only needs to be diagonal. Also, according to Table 2, we find that low-rank projector setting consistently improves the performance, which verifies proposition 2.

**Linear probe on subvector instead of the entire vector**: For *DirectCLR*, we perform a linear probe only on the sub-vector **z** and get $47.9\%$ accuracy on ImageNet. This shows that the rest of **r** still contains useful information even though it does not see gradient directly coming from the loss function.

**Random dropout instead of fixed subvector**: Since *DirectCLR* drops out a number of dimensions for the loss function, it would be natural to ask whether random dropping out can reach the same performance. We train a SimCLR model without a projector and randomly feed $d_0$ number of features to InfoNCE loss every iteration. This model reaches only $43.0\%$ accuracy on ImageNet. This demonstrates the importance of applying a fixed subvector, which allows the alignment effect to happen.

