# OpenReview forum: "Understanding Dimensional Collapse in Contrastive Self-supervised Learning"
_ICLR.cc/2022/Conference — ICLR 2022 Poster_

### Official Review · Reviewer_9VqB · 2021-11-01

**Correctness:** 3
**Technical Novelty And Significance:** 3
**Empirical Novelty And Significance:** 2
**Recommendation:** 6
**Confidence:** 4

**Main Review:**

Strengths:
1. The paper focuses on the dimensional collapse problem faced by contrastive learning methods, which is a valuable problem to be discussed.
2. Authors analyze the collapsed problem by analyzing the magnitude of different singular values of the covariance matrix of the embeddings. Furthermore, they analyze the two reasons which may cause the dimensional collapse. The analysis in toy model (e.g. linear layer, additive noise) is convincing.

Weakness:
1. In sec.4, authors analyze the dynamics of the gradient flow with a linear model, and conclude that strong augmentation will derive low-rank covariance matrix, through lemma 2. However, the analysis lacks experimental validations. Authors do not answer the question that what kinds of augmentations are "strong", and only show the toy model that augmentation is additive noises. I believe the question is important to the contrastive learning.
2. It seems that the analysis in Sec 5 is the properties of all deep models? So I am puzzled that if the Theorem 3 only means the gradients vanishing problem in deep models, I am looking forward to the reply of authors to give a more clear explanation.
3. About the DirectCLR, I thought d_0 is an important hyper-parameters. However there lacks the ablations on d_0.

**Summary Of The Paper:**

The paper discusses the dimensional collapsed problem in contrastive learning, and analyze two reasons to cause the dimensional collapses solutions. According to the analysis, they provide a simple method named DirectCLR to solve the dimensional collapse problem.

**Summary Of The Review:**

The analysis is insightful for some toy settings, but lacks some extensions to the real settings in Sec4. Also experiments are not adequent to validate the points.

---

> ### Author Response · Authors · 2021-11-17
> **Response to Reviewer 9VqB**
>
> > “In sec.4, authors analyze the dynamics of the gradient flow with a linear model, and conclude that strong augmentation will derive low-rank covariance matrix, through lemma 2. However, the analysis lacks experimental validations.
>
> Thanks for the suggestion. We added an experiment to verify the theory in Section 4. Please see results in “Reply to All Reviewers” Q3, which clearly verifies the collapsing effect with an increasing augmentation amplitude.
>
>
> > Authors do not answer the question that what kinds of augmentations are "strong", and only show the toy model that augmentation is additive noises. I believe the question is important to the contrastive learning.”
>
>
> Sorry for the confusion. By “strong augmentation”, we simply mean that the amplitude of the additive augmentation is large. We do not mean more complex augmentation in this linear setting.
>
> Thus, for more complex nonlinear networks, the collapsing condition will depend on the more complicated property of the augmentation (higher-order statistics of augmentation, manifold property of augmentation vs. data distribution) as well as the capacity of the networks.
> We argue that a deep model will not always find an optimal solution if (1) the model has limited capacity or (2) the augmentation is strong in amplitude that surpasses the data distribution.
> It may be true that our standard CNN encoder is powerful enough to find this optimal solution. Then it may suggest that Case 2 is the reason we observe collapse in real SimCLR models.
>
>
> Please also check out more detailed discussion regarding “strong augmentation” in “Reply to All Reviewers” Q3.
>
>
>
> > “It seems that the analysis in Sec 5 is the properties of all deep models? So I am puzzled that if the Theorem 3 only means the gradients vanishing problem in deep models, I am looking forward to the reply of authors to give a more clear explanation.”
>
> Thanks for raising this important question.
>
> The key difference between Theorem 3 and the standard gradient vanishing problem is that the layers evolve to be aligned under Theorem 3’s condition.
>
> In standard gradient vanishing problems, when the gradient matrix propagates through deep layers, its singular values become smaller with deeper layers. This is the effect of a large number of random matrix multiplications. So it requires deep layers to observe vanishing dimensions.
>
> On the contrary, in theorem 3, the collapsed dimensions are specifically aligned together so that the singular values co-evolve over time, and some of them grow much slower than others. Therefore, it does not require “deep networks” to observe vanishing dimensions.
>
> The fundamental difference is that, in theorem 3, the X matrix is positive semidefinite, which is not satisfied in general supervised learning settings.
>
> Therefore, the dimensional collapsing effect caused by implicit regularization is not a general effect in deep models. It is a unique property of contrastive learning.
>
>
>
> > “About the DirectCLR, I thought d_0 is an important hyper-parameters. However there lacks the ablations on d_0.”
>
> We thank the reviewer for the suggestion. We added the detailed hyperparameter tuning on $d_0$ in **Appendix E, Table 3**.
> Please also find this hyperparameter tuning result in “Reply to All Reviewers” Q6.

---

### Official Review · Reviewer_fU6b · 2021-11-02

**Correctness:** 2
**Technical Novelty And Significance:** 3
**Empirical Novelty And Significance:** 2
**Recommendation:** 6
**Confidence:** 3

**Main Review:**

### Strengths

* The mathematical analysis of the representation space is insightful.
* The claims are theoretically well described and the process of proof is easy to follow with the supplementary proofs in the appendix.


### Weaknesses

* No empirical results regarding the strength of augmentation

  Though it is mathematically claimed that the strong augmentation evokes the dimensional collapse of the embedding space, there is no such experimental results that advocate this idea. Besides, how can we find the strength of augmentation that results in the row-rank covariance matrix?

* The gap between the simple linear matrix and the deep model

  Is it reasonable to make analogy between the raw data in the proof with the features extracted from the convolutional network? Wouldn't deep models (CNN encoder) find an optimal way to differentiate two augmented samples in the training process?

* Lack of analysis of the results

  * It seems that the residual connection has the strongest impact on the result, so I feel that more analysis on this factor is needed. Such as,
       * How this  result interpreted with one of the claims?
       * How about the result when the residual connection is applied on SimCLR?
  * There is no explicit advantage of proposed method in terms of the classification performance and the conservation of ranks of the covariance matrix. If SimCLR-based models with projectors are doing well, what is the main advantage of the proposed model?

* Lack of details

  * How do you decide the values of d_0 for the contrastive loss?
  * What is the dimension of the h, z and r in the Figure 10?
  * Specification of Conv Block in the Figure 10?




**Summary Of The Paper:**

This paper theoretically proves that the contrastive learning results in the dimensional collapse in the feature representation space.
By analyzing the covariance matrix of the embedding space, the authors claims that it is specifically arise from the strong augmentation and the implicit regularization.
To circumvent this issue, the authors propose a novel method of contrastive learning without using the projector at the end of an encoder.

**Summary Of The Review:**

The theoretical analysis of the embedding space are insightful, but it seems not well connected with the experimental results.
Also, the gap between the model in the proof process and the ones in the experiment part are quite large.
I think this paper need some additional work on the experimental part.

---

> ### Author Response · Authors · 2021-11-17
> **Response to Reviewer fU6b (part 1 / 2)**
>
> > “No empirical results regarding the strength of augmentation
> Though it is mathematically claimed that the strong augmentation evokes the dimensional collapse of the embedding space, there is no such experimental results that advocate this idea. Besides, how can we find the strength of augmentation that results in the row-rank covariance matrix?”
>
> Thanks for the suggestion.
> Please find our detailed answer to this question in “Reply to All Reviewers” Q1, along with the experimental results advocating the theory in Sec 4.
>
>
> > “The gap between the simple linear matrix and the deep model
> Is it reasonable to make analogy between the raw data in the proof with the features extracted from the convolutional network? Wouldn't deep models (CNN encoder) find an optimal way to differentiate two augmented samples in the training process?”
>
>
> Thanks for raising this important question. For linear settings, “strong augmentation” solely depends on the condition of whether $X$ is positive semidefinite. For more complex nonlinear networks, the collapsing condition will depend on the more complicated property of the augmentation (higher-order statistics of augmentation, manifold property of augmentation vs. data distribution) as well as the capacity of the networks.
> We argue that a deep model will not always find an optimal solution if (1) the model has limited capacity or (2) the augmentation is strong in amplitude that surpasses the data distribution.
> It may be true that our standard CNN encoder is powerful enough to find this optimal solution. Then it may suggest that Case 2 is the reason we observe collapse in real SimCLR models.
>
>
> > “It seems that the residual connection has the strongest impact on the result, so I feel that more analysis on this factor is needed. Such as,
> How this result interpreted with one of the claims?
> How about the result when the residual connection is applied on SimCLR?”
>
>
> Our current theory was focused on linear networks, and empirical results (Figure 6) show that the theory also holds in with nonlinearity: deeper networks have more collapsed dimensions.
>
> For simple nonlinear deep networks, we are supposed to observe dimension collapse in every layer in the network, including both the encoder and the projector. However, the fact is that with ResNet50 backbone, the collapse stops at the representation space. Therefore, there are more complicated dynamics happens with residual connection. Our Figure 10 provides limited understanding on why there’s no collapsed dimensions in representation space. To study the specific dynamics within residual connection will be our future work.
>
> Therefore, our experiment on ImageNet only focuses on the projector whose dynamics matches our theory well.
>
> We want to clarify that the residual connection is not “applied” to the model, and it’s the original building block in the ResNet50 backbone. Standard SimCLR backbone already contains residual connection, and we suspect that’s why the projector can prevent dimension collapse in the representation space.
>
>
>
> > “There is no explicit advantage of proposed method in terms of the classification performance and the conservation of ranks of the covariance matrix. If SimCLR-based models with projectors are doing well, what is the main advantage of the proposed model?”
>
> Please find our detailed answer to this question in “Reply to All Reviewers” Q4.
>
>
> > “How do you decide the values of d_0 for the contrastive loss?”
>
> We decide $d_0$ by a hyperparameter search. Please find our detailed answer to this question in “Reply to All Reviewers” Q6.

---

> > ### Author Response · Authors · 2021-11-17
> > **Response to Reviewer fU6b (part 2 / 2)**
> >
> > > “What is the dimension of the h, z and r in the Figure 10?”
> >
> > Thank you very much for pointing out the lack of clarity. This is very helpful to improve our paper. We provided all the details below and updated the paper accordingly.
> >
> > Figure 10 is an illustrative figure to explain why the whole representation vector is meaningful. We only plot the channel dimension. The dimension of `h`, `r`, `z` depends on the backbone network and the $d_0$ hyperparameter in our model. Therefore, `h` has a dimension `[channel, width, width]`, `r` has a dimension `[channel]`, and `z` has a dimension `[d_0]`.
> >
> > Specifically, for the ResNet50 backbone and the best $d_0$ value, `h` has a dimension `[2048, 7, 7]`, `r` has a dimension `[2048]`, `z` has a dimension `[360]`.
> >
> > We also added the implementation details in the **Appendix D**.
> >
> > > “Specification of Conv Block in the Figure 10?”
> >
> > The Conv Block in Figure 10 is just the last block of ResNet50. Specifically, it is defined by:
> > ```
> > Bottleneck(
> >       (conv1): Conv2d(2048, 512, kernel_size=(1, 1), stride=(1, 1), bias=False)
> >       (bn1): BatchNorm2d(512, eps=1e-05, momentum=0.1, affine=True, track_running_stats=True)
> >       (conv2): Conv2d(512, 512, kernel_size=(3, 3), stride=(1, 1), padding=(1, 1), bias=False)
> >       (bn2): BatchNorm2d(512, eps=1e-05, momentum=0.1, affine=True, track_running_stats=True)
> >       (conv3): Conv2d(512, 2048, kernel_size=(1, 1), stride=(1, 1), bias=False)
> >       (bn3): BatchNorm2d(2048, eps=1e-05, momentum=0.1, affine=True, track_running_stats=True)
> >       (relu): ReLU(inplace=True)
> > ```
> > Note that we only plotted the channel dimension and omitted the averagepool.
> > ```
> > (avgpool): AdaptiveAvgPool2d(output_size=(1, 1))
> > ```

---

> ### Author Response · Authors · 2021-11-26
> **Has our response addressed your concerns?**
>
> Dear Reviewer fU6b:
>
> We would be grateful if you can confirm whether our response has addressed your concerns and let us know if any issues remain. In the following, we recap the key points of our response:
>
> 1. We provided additional experiments on “strong augmentation” which perfectly matches our theory.
> 2. We added more empirical experiments on “implicit regularization” on nonlinear settings in Figure 6. This verifies that our theory also applies to deep nonlinear networks.
> 3. We provided extra analysis and ablation studies, which better explain how our proposed model is connected to the theory. The propositions are all verified by our ablation study.
> 4. We added all experiment details, including (a) detailed hyperparameter tuning on $d_0$ (b) more details in dimensionality for Figure 10 which explains why `z[d_0: ]` in DirectCLR is not trivial (c) Network setting details of the experiments.
>
> We are looking forward to your feedback!

---

> > ### Comment · Reviewer_fU6b · 2021-11-27
> > **Response to Authors**
> >
> > Thanks for the detailed comments and efforts on the additional experimental results.
> > It seems that the newly added results well support the theoretical analysis.
> > So I raised a score for that.
> >
> > But I still feel that my concern remains in that the claims are not well connected throughout the paper.
> > For me, It seems like the authors try to solve the problem of dimensional collapse until Section 5.
> > So, the claim until this point is that the strong augmentation and the implicit regularization are the factors that dampen the training process.
> >
> > But it is not well connected to the result of SimCLR.
> > First, (i) there is no analysis on the strength of the augmentation for SimCLR, and (ii) the result in Table 1 doesn't really coincide with the claim, since it implies that the more layers and nonlinearity in the projector, the better the performance.
> > Moreover, I checked the authors' comment regarding Q1, but I still can't find the meaning of the proposed method, DirectCLR.
> > If the low rank solution is the problem to solve, shouldn't the method handle the problem by rendering the model to have a higher rank? Why would you stick to the phenomenon?
> >
> > In sum, the theoretical analysis and the added empirical results are impressive, but I feel there an explicit logical instability from Section 6, which I believe it should be improved.

---

> > > ### Author Response · Authors · 2021-12-02
> > > **Response to Reviewer fU6b**
> > >
> > > (We are sorry we posted the wrong response just now. It was for another reviewer. Here's the correct response.)
> > >
> > > We thank the reviewer for being super insightful and detailed about the paper. We really appreciate the feedback!
> > > We are glad to hear that the reviewer likes our theoretical analysis with supporting empirical evidence. On the other hand, we find that the remaining confusion is still how the proposed model is connected to the theory.
> > >
> > > First, we want to emphasize that the proposed model is **not aimed to improve the state-of-the-art** but to **explain the internal dynamics of contrastive learning**. The empirical advantage of our proposed model is that we can remove those parameters in the projector without affecting the performance. This empirical advantage is not the main focus of the paper and we will downtone it.
> > >
> > > Second, we believe there is confusion on the "embedding space" vs. "representation space". This is the standard setting in SimCLR that we don't use the final collapsed "embedding space" for downstream tasks but the layer before the projector.
> > >
> > > Here are our detailed answers to your concerns:
> > >
> > > > there is no analysis on the strength of the augmentation for SimCLR
> > >
> > > Thanks for the suggestion. This is very helpful for us to improve the scope of the paper.
> > >  We performed extra experiments showing that with various augmentation values (by tuning the augmentation parameters such as cropping, color jittering), SimCLR will always suffer from dimensional collapse. This perfectly matches our theory that the dimensional collapse will happen in both weak and strong augmentation settings caused by different mechanisms. We will add these experiment results in our final paper.
> > >
> > > > the result in Table 1 doesn't really coincide with the claim since it implies that the more layers and nonlinearity in the projector, the better the performance
> > >
> > > We believe this is a confusion, which is also raised by Reviewer HfRQ.
> > >
> > > The SimCLR model does not use final embedding space for downstream tasks. The performance is based on representation vectors, which is before the projector.
> > > More layers in the projector truly result in more collapsed dimensions in embedding space. However, it does not mean the hidden layer, which is a few layers behind, will also suffer from it.
> > > That's why a 2-layer projector SimCLR performs better than a single linear layer projector. In fact, the performance will not increase if an even deeper projector is used.
> > >
> > >
> > > > but I still can't find the meaning of the proposed method, DirectCLR. If the low-rank solution is the problem to solve, shouldn't the method handle the problem by rendering the model to have a gradient with a higher rank? Why would you stick to the phenomenon?
> > >
> > > The gradient coming from the InfoNCE is full rank. Our theory explains why this gradient will drive the space into low-rank.
> > > Also, we stick to the phenomenon because it is exactly what happened in the SimCLR projector. The goal of the proposed model is to show our theory can explain the phenomenon.
> > > In fact, there are many other approaches to solve this problem e.g. [3, 4].
> > >
> > > [3] Aleksandr Ermolov, Aliaksandr Siarohin, Enver Sangineto, Nicu Sebe, “WMSE: Whitening for Self-Supervised Representation Learning”, ICML’21
> > >
> > > [4] Jure Zbontar, Li Jing, Ishan Misra, Yann LeCun, Stéphane Deny, “Barlow Twins: Self-Supervised Learning via Redundancy Reduction”, ICML’21

---

> > > > ### Comment · Reviewer_fU6b · 2021-12-02
> > > > **Response to Authors**
> > > >
> > > > Thanks for the additional explanation. I think it clearly answered some of my questions.
> > > > I raised a score again anticipating the further enhanced paper.

---

### Official Review · Reviewer_t7MP · 2021-11-02

**Correctness:** 4
**Technical Novelty And Significance:** 4
**Empirical Novelty And Significance:** 4
**Recommendation:** 8
**Confidence:** 3

**Main Review:**

### Strength

The paper presents and studies an important problem in all joint-embedding, comparison-based, self-supervised learning techniques: how to avoid degenerative solutions when representations are collapsed. This problem was previously widely known for the “completely collapse” case, where every representation is mapped to a single point, hence the introduction of explicit negative pairs in contrastive methods to avoid this trivial solution. Recently, some methods have been shown to be able to learn without explicitly negative samples, but these methods suffer from a milder collapsing problem, only spanning a subspace of the given capacity. This paper observes that naive contrastive methods without an embedding projector also suffer from this “dimension collapse” problem. The motivation, problem and current state of understanding are clearly described in Section 2, 3 and intuitively illustrated in Fig.1 and Fig.2.

By analyzing the gradient flows in a simplified setting with shallow and linear network, Section 4 and 5 prove two different scenarios of why some dimension collapse and embedding space converge to low-rank solution. Both analysis maps well to intuition regarding the problem.
- The cause in Section 4 can be intuitively understood as: when augmentation is too strong for a given model capacity, that destroys too much similarity information, the representations span a subspace of the full capacity to keep the similarity score high.
- The causes in Section 5 are studied from the implicit regularization point of view, where multi-layers (linear) networks are implicitly biased to learn a low rank solution.

The projector is an important heuristic design choice of many state-of-the-art contrastive methods, with or without using explicit negative pairs. Previously the projector component was only motivated by its computation efficiency when computing similarity scores in a lower-dimensional space, however there was no explanation for the performance loss without it. In the light of the dimensionality collapse phenomenon, the projector now shields the dimension of the representation layer from collapsing, thus allowing it to retain more useful information for transfer tasks.

### Concerns

The theoretical analysis in Section 4 and 5 both relies on the linearity property in the simplified setting. Especially in section 5, where the study of weights alignment seems to only be applicable between two linear layers. The relevance to modern deep nonlinear networks are made only through some empirical evidence from section 5.4. This seems to be a general limitation in the toolbox for theoretical analysis as a field, not just for this paper, so in Figure 6, it would be nice to see more empirical evidence (plots) for multi-layer networks with non-linearity at the same time.

The proposed DirectCLR training method is equivalent to 1-layer linear projector (SimCLR v1 [1]), so it only explained some benefits of the projector components. In case of multi-layer non-linear projector (SimCLRv2 [2]), dimensionality collapse as visualized using singular value in Fig.9 can not explain the performance gain completely. While this was shown in the first line of Table 1, it was not discussed anywhere in the text. A small discussion of other potential effects of non-linear projector (see review paper [3]) for future research would be helpful.

I am a little confused about Fig.10 and its caption. The caption says “The rest part the representation ... is copied from the previous layer via residual connection, which experiences full rank gradient passed through the last convolution block.” However in the figure, the gradient vector through the residual connection was annotated with “low-rank”. It would be great if the paragraph before Fig.10 can be made better to understand.


**Summary Of The Paper:**

This paper shows that contrastive methods also suffer from the “dimensionality collapse” phenomenon, a milder version of the “total collapse” problem that originally motivated the development of using explicit negative pairs to prevent learning trivial solutions. Two underlying causes for this problem, i.e too strong data augmentation and implicit low-rank regularization, are proved in simplified settings of shallow, linear networks. The dimensionality collapse problem is then related to the embedding projector component of SimCLR, and theoretical analyses are used to motivate an alternative training technique without the need for the embedding projector.


**Summary Of The Review:**

This paper empirically demonstrates and theoretically investigates an important problem of representation learning in general and contrastive methods in particular. The paper provides useful insights, related it to a current heuristic architectural design (projection layers) and proposes a novel training technique to get rid of that heuristic.
While the theoretical analysis is performed under simplified settings and its applicability to deep networks are only shown empirically, its limitation is inline with other theoretical works of the field.
For the reason above I believe this is a useful contribution to the field and hope to see this get accepted at ICLR.

##### References
[1] Chen, Ting, et al. "A simple framework for contrastive learning of visual representations." International conference on machine learning. PMLR, 2020.

[2] Chen, Ting, et al. "Big self-supervised models are strong semi-supervised learners." arXiv preprint arXiv:2006.10029 (2020).

[3] Le-Khac, Phuc H., Graham Healy, and Alan F. Smeaton. "Contrastive representation learning: A framework and review." IEEE Access (2020).

---

> ### Author Response · Authors · 2021-11-17
> **Response to Reviewer t7MP**
>
> > “The theoretical analysis in Section 4 and 5 both relies on the linearity property in the simplified setting. Especially in section 5, where the study of weights alignment seems to only be applicable between two linear layers. The relevance to modern deep nonlinear networks are made only through some empirical evidence from section 5.4. This seems to be a general limitation in the toolbox for theoretical analysis as a field, not just for this paper, so in Figure 6, it would be nice to see more empirical evidence (plots) for multi-layer networks with non-linearity at the same time.”
>
> Thanks for the suggestion. We added more empirical experiments regarding multilayer-nonlinear experiments. See updated **Figure 6** in the paper.
>
> We observe that a similar phenomenon holds for the nonlinear case: deeper layers have stronger collapsing effects. The singular value spectrum seems to have a more complicated distribution for nonlinear cases compared to linear ones where the singular values seem to form groups. We will explore the nonlinear dynamics theory in the future.
>
>
> > “The proposed DirectCLR training method is equivalent to 1-layer linear projector (SimCLR v1 [1]), so it only explained some benefits of the projector components. In case of multi-layer non-linear projector (SimCLRv2 [2]), dimensionality collapse as visualized using singular value in Fig.9 can not explain the performance gain completely. While this was shown in the first line of Table 1, it was not discussed anywhere in the text. A small discussion of other potential effects of non-linear projector (see review paper [3]) for future research would be helpful.”
>
> Thank you for pointing out the lack of discussion on nonlinear projector effects. Thus, our theory only explains the dynamics behind linear network and hence linear projector SimCLR setting.
>
> Also thanks for suggesting the references. Thus, the SimCLR v2 paper [2] and the review paper [3] have discussed the empirical advantage of the nonlinear projector. The theoretical understanding of the role of nonlinear projectors in contrastive learning is still missing.
>
> We suspect that a nonlinear projector is able to leverage higher order statistics from the data/augmentation which allows dimensions collapsed in linear setting to survive.
>
> This will be an interesting direction to explore in the future.
>
>
>
> > “I am a little confused about Fig.10 and its caption. The caption says “The rest part the representation ... is copied from the previous layer via residual connection, which experiences full rank gradient passed through the last convolution block.” However in the figure, the gradient vector through the residual connection was annotated with “low-rank”. It would be great if the paragraph before Fig.10 can be made better to understand.”
>
>
> Thank you very much for pointing out the lack of clarity in our paper. This is really helpful to improve our paper. Here, we provided all the details below and updated the paper accordingly.
>
> Because the rest of the representation vector `r[d_0:]` does not see gradients from the InfoNCE loss. Intuitively, this part of the vector would not contain any useful information.
>
> This paragraph aims to explain that due to the residual connection in the backbone, this part of the representation vector is not trivial.
>
> The gradient passing through the representation vector ($dL/dv$) only applies to the first $d_0$ dimensions. We omitted the averagepool layer because it is applied channel-wise independently. Therefore, when gradient backpropogates towards the conv block, it is still low-rank, with only the first $d_0$ dimension non-zero. However, when this gradient backpropogates through the conv block, it becomes full rank and applies to all channel dimensions. Therefore, the previous layer $h$ is fully trained, and every channel in $h$ is meaningful. Note that we omitted the spatial dimension and only focused on the channel dimension. Given that $h$ is full-rank and every channel in $h$ is meaningful, we know that the residual output: `conv(h) + h` is also full-rank. This means that $r$ has all channels containing useful information.
>
> Sorry for the confusion on the “low-rank” annotation on the residual connection. This means that the hidden layer $h$ observes two gradients, one from residual connection, one from the conv block. The gradient from the residual path is low-rank, but since the gradient from the conv block is already full-rank, we know in total the hidden layer $h$ experiences a full-rank gradient.
>
> We updated **Figure 10** and the paragraph in the paper with a clearer explanation.

---

> > ### Comment · Reviewer_t7MP · 2021-11-30
> > **Response to authors**
> >
> > Thanks for improving the paper and addressing my concerns.
> > Having read the the updated version, I retain my original evaluation and recommend for accept.
> >
> > This is an interesting paper that shed light on some inner working of Contrastive method. I personally like these type of papers more than paper chasing SOTA. I suspect if the paper does not emphasize too much on the propose DirectCLR but position itself more of an investigative paper, that would make the message clearer.
> >
> > After all, I agree with other reviewers that selling"DirectCLR,  a novel algorithm outperforms SimCLR with linear projector" does not make much of an impression, since empirically extending SimCLR to non-linear projector is trivial, but extending the theoretical analysis of DirectCLR to non-linear projector is not.

---

> > > ### Author Response · Authors · 2021-12-01
> > > **Response to Reviewer t7MP**
> > >
> > > We thank the reviewer for being supportive of our paper. We agree with the reviewer that the proposed method should not be emphasized too much as the only purpose of this model is to justify the theory. We will downtone the claims on "DirectCLR, a novel algorithm outperforms SimCLR with linear projector" and adjust the emphasis in our final version.

---

### Official Review · Reviewer_HfRQ · 2021-11-02

**Correctness:** 3
**Technical Novelty And Significance:** 3
**Empirical Novelty And Significance:** 3
**Recommendation:** 6
**Confidence:** 4

**Main Review:**

This paper is interesting. The proposed DirectCLR dropping projector is a novel idea. The experiments are also exciting, which shows that the projector may not be necessary for contrastive learning. I would like to vote for a “weak accept”, because I also have the following concerns:

1). The motivation is not completely solid. I agree that dimensional collapse is an essential issue that should be avoided in training. However, directly increasing the dimensionality is also not good for the model generalization, which is usually regarded as the curse of dimensionality. Sometimes, the low-rank/low-dimensional data may help us find out the intrinsic low-dimensional manifold, and that is why we need the dimensionality reduction technique.

2). The proposed method seems to be contrary to motivation. If we want to avoid dimensional collapse, why do we further reduce the fitting ability of the network? Do fewer features incur the more heavily dimensional collapse?

3). Directly selecting the features from 0 to d0 is simple, but how can we ensure that the useful features are just located in [0, d0]?

**Summary Of The Paper:**

This paper firstly studies the dimensional collapse problem in existing contrastive learning (CL) methods. The authors provided both empirical and theoretical results to reveal that the popular SimCLR method may incur dimensional collapse by showing the low-rank property of the covariance. They thus propose a new framework called DirectCLR to solve the dimensional collapse in CL. Experiments on ImageNet demonstrate the effectiveness of the proposed method.


**Summary Of The Review:**

See above.

---

> ### Author Response · Authors · 2021-11-17
> **Response to Reviewer HfRQ**
>
> > “1). The motivation is not completely solid. I agree that dimensional collapse is an essential issue that should be avoided in training. However, directly increasing the dimensionality is also not good for the model generalization, which is usually regarded as the curse of dimensionality. Sometimes, the low-rank/low-dimensional data may help us find out the intrinsic low-dimensional manifold, and that is why we need the dimensionality reduction technique.”
>
> We thank the reviewer for raising this important question. Please see “Reply to All Reviewers” Q1 and Q4 for our detailed discussion on motivation.
>
> We agree that using a low-dimensional representation is the goal of representation learning and essential for the model generalization ability. There may exist an optimal dimensionality or intrinsic dimension manifold for the dataset. But the current dimensionality in the contrastive learning framework is lower than the intrinsic dimension in ImageNet with ResNet. One evidence is that increasing the dimensionality (with a WideResNet) can improve the performance. Therefore, the dimensional collapse in the contrastive learning model is not ideal. Considering an extreme scenario, an infinite powerful model will only try to use minimal dimensions to distinguish between images. Then the model will not learn enough useful information in the representations for downstream tasks.
>
> In addition, our empirical results in Figure 9 and Table 1 show that dimensional collapse in contrastive learning harms performance. Therefore, additional empirical motivation is to avoid such collapse to learn better representation.
>
>
> > “2). The proposed method seems to be contrary to motivation. If we want to avoid dimensional collapse, why do we further reduce the fitting ability of the network? Do fewer features incur the more heavily dimensional collapse?”
>
> Thanks for this question. According to our theory and propositions, many of the projector parameters are not necessary. (See more discussion in “Reply to All Reviewers Q1”. For example, the orthogonal component in the weight matrix is redundant as it will evolve to align with the previous layers. The singular values also only need to be fixed. DirectCLR with these settings can have the same effect of “avoiding dimensional collapse” as a projector. That’s why we can simplify the model without affecting the fitting ability.
>
>
> > “3). Directly selecting the features from 0 to d0 is simple, but how can we ensure that the useful features are just located in [0, d0]?”
>
> Please find our detailed answer to this question in “Reply to All Reviewers” Q2.

---

### Official Review · Reviewer_z9oW · 2021-11-03

**Correctness:** 3
**Technical Novelty And Significance:** 2
**Empirical Novelty And Significance:** 3
**Recommendation:** 5
**Confidence:** 4

**Main Review:**

In 4.2 GRADIENT FLOW DYNAMICS, what is the meaning of $X$ in Eq. 6? How amplitude of augmentation is reflected in Eq. 7? The proof is based on simple linear network setting. I wonder how it extends to more complex nonlinear networks. Also, how 'strong’ is strong augmentation? With more complex structures, the capacity of the network also increases, and I expect it would be more tolerable to `strong’ augmentation.

The propositions regarding the role of projector are lack of justification. I am not sure whether the claims are correct or not.

DirectCLR picks a subvector of the representation z = r[0 : d_0]. Is there any assumption on the order of feature dimensions? Why do we expect the subvector based approach helps? How it compares to random dropout of features?

The experimental validation is quite limited. There are a few recent work addressing the problem and propose to leverage only positive pairs for learning. The authors may want to compare with them.
[1] BYOL: Bootstrap Your Own Latent A New Approach to Self-Supervised Learning, NIPS 2020
[2] SimSiam: Exploring Simple Siamese Representation Learning, CVPR 2021
[3] WMSE: Whitening for Self-Supervised Representation Learning, ICML 2021

**Summary Of The Paper:**

This paper investigates the collapsing problem of contrastive learning. It attempts to attribute the collapsing phenomena to strong augmentation and implicit regularization, using simple linear network models. Based on the above analysis, the paper then propose a simple sub-vector based CL method called DirectCLR.

**Summary Of The Review:**

This paper investigate an important and interesting question for CL. The analysis performed is interesting but more justification may be needed for some propositions. However, the experimental validation is quite limited.

---

> ### Author Response · Authors · 2021-11-17
> **Response to Reviewer z9oW (part 1 / 2)**
>
> > “In 4.2 GRADIENT FLOW DYNAMICS, what is the meaning of X  in Eq. 6?”
>
> According to *Lemma 2*, $X$ is defined by the weighted “data distribution” covariance matrix subtracting the weighted “augmentation distribution” covariance matrix.  According to *Lemma 1*, the gradient on weight matrix W is proportional to $X$.
>
> Therefore, whether the gradient is rank-deficient depends on whether X is positive semidefinite. If $X$ is not positive semidefinite,  we will observe that $W$ converges to low-rank.
>
> A high-level understanding of $X$ is that when augmentation is stronger than data distribution along a particular dimension, the model will collapse along the corresponding dimension.
>
> .
> >  “How amplitude of augmentation is reflected in Eq. 7?”
>
> In Eq.7, $X = \hat{\Sigma}_0 - \hat{\Sigma}_1$. Here $ \hat{\Sigma}_1$ is a weighted version of “amplitude of augmentation”. The amplitude of augmentation is characterized by the covariance matrix of the vector $x_i’-x_i$. So stronger augmentation means larger $ \hat{\Sigma}_1$ and therefore $X$ has more negative eigenvalues.
>
> Please find our detailed answer to this question in “Reply to All Reviewers” Q3.
>
> > “The proof is based on simple linear network setting. I wonder how it extends to more complex nonlinear networks. Also, how 'strong’ is strong augmentation? With more complex structures, the capacity of the network also increases, and I expect it would be more tolerable to `strong’ augmentation.”
>
> Sorry for the confusion. By “strong augmentation”, we simply mean that the amplitude of the additive augmentation is large. We do not mean more complex augmentation in this linear setting.
>
> For more complex nonlinear networks, the collapsing condition will depend on the more complicated property of the augmentation (higher-order statistics of augmentation, manifold property of augmentation vs. data distribution) as well as the capacity of the networks.
>
> Please find our detailed answer to this question in “Reply to All Reviewers” Q3.
>
>
> > “The propositions regarding the role of projector are lack of justification. I am not sure whether the claims are correct or not.”
>
> Thanks for raising this important question.
> Please find our detailed answer to this question in “Reply to All Reviewers” Q1.
>
> > “DirectCLR picks a subvector of the representation z = r[0 : d_0]. Is there any assumption on the order of feature dimensions?”
>
> We claim that the order of the features does not matter. As long as you have picked a number fixed number of dimensions, it will give the same performance.
>
> The reason is the alignment effect. The orthogonal component of the projector weight matrix will evolve to align to the previous layer according to our theory. The singular values in the weight matrices will always be paired to the previous layer. Therefore, changing the order will not result in any difference.
>
> This is also explained in detail in “Reply to All Reviewers” Q2.
>
> In addition, we added an ablation study of picking a number of FIXED dimensions for the subvector.
> ```
> self.indices = np.random.choice(2048, d_0, replace=False) # fixed throughout training
> ...
> z = r[self.indices]
> ```
> It performs exactly the same as doing `z = r[0 : d_0]`.
>
>
>
> > Why do we expect the subvector based approach helps?
>
> Please find our detailed answer to this question in “Reply to All Reviewers” Q4.
>
>
>
> > “How it compares to random dropout of features?”
>
> Thanks for suggesting this comparison. Here, we run an ablation study using random dropout of features, i.e., each iteration, the subvector indices are different.
> ```
> indices = np.random.choice(2048, d_0, replace=False) # every iteration is different
> z = r[indices]
> ```
> This model only gives 43.0% accuracy on ImageNet for 100 epoch pretraining, which is even worse than the baseline model (SimCLR without projector gives 51.5%).
>
> It’s easy to understand that a random dropout is effectively equivalent to no projector but much noisy. All dimensions are directly exposed to the InfoNCE loss. This actually results in more dimensional collapse.
>
> We also added this ablation study to the paper in **Appendix F**.

---

> > ### Author Response · Authors · 2021-11-17
> > **Response to Reviewer z9oW (part 2 / 2)**
> >
> > > "The experimental validation is quite limited. There are a few recent work addressing the problem and propose to leverage only positive pairs for learning. The authors may want to compare with them. [1] BYOL: Bootstrap Your Own Latent A New Approach to Self-Supervised Learning, NIPS 2020 [2] SimSiam: Exploring Simple Siamese Representation Learning, CVPR 2021 [3] WMSE: Whitening for Self-Supervised Representation Learning, ICML 2021"
> >
> > Thanks for the suggestion. We added these papers to our discussion.
> >
> > The goal of our theory is to understand the counterintuitive phenomenon of dimensional collapse in contrastive learning rather than just solving it by patching the loss function.
> >
> > The dimensional collapse in non-contrastive learning methods like BYOL and SimSiam is easier to understand. There’s no negative term to prevent such collapse. More detailed dynamics have already been studied in [6].
> >
> > Whitening methods like [3] WMSE and [4, 5] are a good approach to solve dimensional collapse directly from the loss function. And this is probably the main reason why this approach performs better than contrastive methods.
> >
> >
> > [4] Jure Zbontar, Li Jing, Ishan Misra, Yann LeCun, Stéphane Deny, “Barlow Twins: Self-Supervised Learning via Redundancy Reduction”, ICML’21
> >
> > [5] Adrien Bardes, Jean Ponce, Yann LeCun, “VICReg: Variance-Invariance-Covariance Regularization for Self-Supervised Learning”, arXiv: 2105.04906
> >
> > [6] Yuandong Tian, Xinlei Chen, Surya Ganguli, “Understanding self-supervised Learning Dynamics without Contrastive Pairs“, ICML’21

---

> ### Author Response · Authors · 2021-11-26
> **Has our response addressed your concerns?**
>
> Dear Reviewer z9oW:
>
> We would be grateful if you can confirm whether our response has addressed your concerns and let us know if any issues remain. In the following, we recap the key points of our response:
>
> 1. We clarified the meaning of strong augmentation. And we have provided additional experiments to verify the prediction of the theory.
> 2. We clarified the connection between the theory and the proposed models. Our additional ablation study justified the propositions. It also explains the specific design in DirectCLR. The question of “comparing to random features” is also addressed by further ablation experiments.
> 3. We discussed the connection between our approach and the other whitening/non-contrastive SSL approaches.
>
> We are looking forward to your feedback!

---

### Author Response · Authors · 2021-11-17
**General Response to All Reviewers (part 1 / 3)**

We would like to thank the reviewers for their constructive suggestions, which allowed us to improve our paper. We appreciate that all reviewers find our theory insightful and novel. We also admit the lack of clarity in connecting the theory and the propositions supporting the proposed model. We believe that we were able to respond in depth to all of the concerns and kindly ask the reviewers to consider increasing their scores. We updated the paper accordingly.

Here, we address a few common questions and demonstrate some important additional experiments. We also specifically respond to each reviewer for detailed questions.

> Q1. How is theoretical analysis connected to our empirical study? Justify the propositions on the projector.

We thank the reviewers for raising this important question. We believe this is the most critical point connecting our theory to experiment.

This is also related to the question by Reviewer z9oW and Reviewer HfRQ on the choice of only top $d_0$ indexed features. Here we give a better explanation of the propositions supporting the idea of DirectCLR.

Based on our theory on implicit regularization dynamics, we expect to see adjacent layers $W_1 (=U_1S_1V_1^T)$ and $W_2 (=U_2S_2V_2^T)$ to be aligned such that the overall dynamics is only governed by their singular values $S_1$ and $S_2$. And the orthogonal matrices $V_2$ and $U_1$ are redundant as they will evolve to $V_2^TU_1 = I$, given $S_1$ and $S_2$.

Now, let’s consider the linear projector SimCLR model and only focus on the channel dimension. $W_1$ is the last layer in the encoder, and $W_2$ is the projector weight matrix. Our proposition claims that for this projector matrix $W_2$, the orthogonal components $V_2$ can be omitted. Because the previous layer $W_1$ is fully trainable, its orthogonal component ($U_1$) will always evolve to satisfy $V_2^TU_1 = I$. Therefore, the final behavior of the projector is only determined by the singular values ($S_2$) of the projector weight matrix.

This motivates proposition 1: the orthogonal component of the weight matrix doesn’t matter. So we can set the projector matrix as a **diagonal** matrix.

Also, according to our theory, the weight matrix will always converge to the low-rank. The singular value diagonal matrix $S_2$ naturally becomes low-rank, so why not just set it **low-rank** directly? This is the motivation of proposition 2. However, to better theoretically understand this proposition will be an interesting future work.

Given these two propositions, we propose DirectCLR, which is effectively using a **low-rank diagonal** projector.

Both propositions are justified by our ablation study. See below.

-------------------
**Table 2**: Ablation study: top-1 accuracies on ImageNet by SimCLR model with different projector settings.

| SimCLR projector setting          | diagonal | low-rank | Top-1 Accuracy on ImageNet |
|-----------------------------------|----------|----------|----------------------------|
| No projector                      |          |          | 51.5%                      |
| Orthogonal projector              |          |          | 52.2%                      |
| Trainable linear projector        |          |          | 61.1%                      |
| Trainable diagonal projector      | yes      |          | 60.2%                      |
| Fixed low-rank projector          |          | yes      | 62.3%                      |
| Fixed low-rank diagonal projector | yes      | yes      | 62.7%                      |
-------------------

**Proposition 1** matches the fact that:
1. An orthogonally constrained projector performs the same as the non-projector setting.
2. Fixed low-rank projector performs the same as a fixed diagonal projector.
3. Trainable linear projector performs the same as a trainable diagonal projector.

**Proposition 2** matches the observation that a low-rank projector has the highest accuracy.


> Q2. Justify why we only pick top d_0 features in DirectCLR.

Reviewer z9oW and Reviewer HfRQ questioned the choice of only top $d_0$ indexed features. Choosing top $d_0$ features is equivalent to selecting any fixed $d_0$ features, which is a direct consequence of our theory and propositions. Please see our better explanation on the propositions above in Q1.

Here, we specifically explain why we can choose just the top $d_0$ indexed subvector. The order of singular values in the projector weight matrix ($S_2$) doesn’t matter because the orthogonal component from the previous layer ($U_1$) will always evolve to satisfy $V_2^TU_1=I$ (with sorted $S_1$ and $S_2$). Therefore, the singular values between $S_1$ and $S_2$ will always evolve to be paired regardless of how they are indexed. This justifies the design of simply picking the top $d_0$ indexed subvector.

---

> ### Author Response · Authors · 2021-11-17
> **General Response to All Reviewers (part 2 / 3)**
>
> > Q3. In “dimensional collapse caused by strong augmentation”, what’s the meaning of strong augmentation?
>
> We thank the reviewers (Reviewer z9oW, Reviewer fU6b and Reviewer 9VqB) for raising this important question. We clarify the definition below alongside an extra experiment.
>
> In a linear setting, “strong augmentation” means the amplitude of the additive augmentation is large. It is defined by the condition that “$X = \hat{\Sigma}_0 - \hat{\Sigma}_1$ has negative eigenvalues”. Since $\hat{\Sigma}_1$ is determined by the amplitude of the augmentation (covariance matrix), strong augmentation results in larger $\hat{\Sigma}_1$ and more negative eigenvalues in $X$.
>
> We also empirically demonstrate that there’s a threshold of the “strong augmentation” for the linear case.
>
> Here, we set the data distribution to be an isotropic Gaussian with covariance matrix as identity: $\sum_{i,j}(x_j-x_i)(x_j-x_i)^T/N = I$.
> We set the augmentation as an additive Gaussian with covariance matrix $\sum_i(x_i’-x_i)(x_i’-x_i)^T/N = blockdiag(0, k * Identity)$. We run the single layer contrastive learning model with InfoNCE loss with various $k$ values. We plot the resulting singular value spectrum of the weight matrix in the **Figure 11** in **Appendix C**.
>
> The experiment shows that large augmentation amplitude $k$ results in dimensional collapse in the weight matrix $W$.
> Note the threshold is not precisely $k=1$ because $\hat{\Sigma}_0$ and $\hat{\Sigma}_1$ are weighted by the factors $\alpha$.
>
> From a high-level point of view, strong augmentation can be simply understood as whether it is stronger than data distribution along any certain dimensions.
>
> For linear settings, “strong augmentation” solely depends on the condition of whether $X$ is positive semidefinite. For more complex nonlinear networks, the collapsing condition will depend on the more complicated properties of the augmentation (higher-order statistics of augmentation, manifold property of augmentation vs. data distribution) as well as the capacity of the networks.
>
> > Q4. What’s the motivation of DirectCLR? SimCLR works well with a projector in practice, what’s the advantage of DirectCLR?
>
> We thank Reviewer z9oW, Reviewer HfRQ and Reviewer fU6b for asking this question. Here, we discuss the motivation and advantages of the proposed model.
>
> The motivation of this paper is to understand this widely-ignored and counter-intuitive dimensional collapse phenomenon in contrastive learning. Specifically, for the proposed model, it serves as a justification of our theory and propositions. See detailed discussion on how our model connected to the theory above in Q1.
>
> Previously, the role of the projector in contrastive learning was mysterious. Our method demonstrates that by understanding the dynamics of this component, we can directly replace it with an analytical form. In fact, the superior performance of the model over the linear projector SimCLR proves the correctness of our propositions and our theory. So far, the empirical advantage is that we can get rid of a linear projector and provide slightly higher performance. Our model is a good alternative to linear projector SimCLR.
>
> Thus, our model has not outperformed a standard SimLCR with a nonlinear projector because its design is based on the theory within a linear setting. We will explore the nonlinear theory and hopefully find a theoretical replacement for nonlinear projectors in the future. And we believe that this future theory will share similar principles as the two collapsing scenarios we discovered in the linear setting.
>
> > Q5. The gap between the linear case and nonlinear/deep networks.
>
> Reviewer z9oW and Reviewer fU6b raised the concern on how to extend our theory to complex deep network cases.
>
> First, we thank the suggestion by Reviewer t7MP. We extended our study empirically by running more experiments on nonlinear-multilayer networks. See Figure 6. Even though theoretically, it's still difficult to analyze the dynamics in nonlinear networks, we suspect that the principle of our theory holds for deep networks.
>
> As mentioned by Reviewer t7MP: “While the theoretical analysis is performed under simplified settings and its applicability to deep networks are only shown empirically, its limitation is inline with other theoretical works of the field.”, we argue that studying complex dynamics in deep nonlinear networks is still a challenge for learning theory research. It’s still not well understood enough for simple supervised learning. For example, the literature on implicit regularization is mostly still limited to linear network settings [8,9]. To extend learning theory under complex training scenarios such as self-supervised learning remains future work for the whole community.
>
> [8] Noam Razin, Nadav Cohen, “Implicit Regularization in Deep Learning May Not Be Explainable by Norms”, NeurIPS’20
>
> [9] Noam Razin, Asaf Maman, Nadav Cohen, “Implicit Regularization in Tensor Factorization”, ICML’21

---

> > ### Author Response · Authors · 2021-11-17
> > **General Response to All Reviewers (part 3 / 3)**
> >
> > > Q6. How to find hyperparameter $d_0$?
> >
> > Reviewer fU6b and Reviewer 9VqB asked how we pick the hyperparameter $d_0$. We thank reviewers for asking this question as this is an important experiment missing in the paper. Here, we added an extensive hyperparameter search on $d_0$. See results below:
> >
> > | dim  | accuracy |
> > |------|----------|
> > | 256  | 60.808   |
> > | 300  | 61.126   |
> > | 340  | 61.944   |
> > | 350  | 62.038   |
> > | 360  | 62.104   |
> > | 370  | 61.768   |
> > | 380  | 61.826   |
> > | 390  | 61.802   |
> > | 420  | 61.856   |
> > | 450  | 61.496   |
> > | 768  | 60.860   |
> > | 1024 | 59.740   |
> > | 2048 | 51.652   |
> >
> > It’s easy to see that when $d_0 \rightarrow 0$, there’s too little gradient information coming from the loss, the performance drops. When $d_0 \rightarrow 2048$, the model converges to standard SimCLR without a projector, which we know suffers from dimensional collapse in representation space.
> >
> > We added this result in the **Appendix E Table 3**.

---

### Decision · Program_Chairs · 2022-01-20

**Decision:**

Accept (Poster)

**Comment:**

The theory and results presented in this paper provide a new method to avoid collapse in contrastive learning.  All but one reviewer recommend acceptance.  The lone negative reviewer is concerned with the limited experiments, but the other reviewers, and the AC, find the experimentation convincing enough to warrant acceptance.